# Generalizable Heuristic Generation Through LLMs with Meta-Optimization

**Yiding Shi**[1], **Jianan Zhou**[1*], **Wen Song**[2*], **Jieyi Bi**[1], **Yaoxin Wu**[4], **Zhiguang Cao**[3], **Jie Zhang**[1]

[1]Nanyang Technological University    [2]Shandong University
[3]Singapore Management University    [4]Eindhoven University of Technology
`yiding002@e.ntu.edu.sg  jianan004@e.ntu.edu.sg`
`wensong@email.sdu.edu.cn  jieyi001@e.ntu.edu.sg`
`y.wu2@tue.nl  zgcao@smu.edu.sg  zhangj@ntu.edu.sg`

## Abstract

Heuristic design with large language models (LLMs) has emerged as a promising approach for tackling combinatorial optimization problems (COPs). However, existing approaches often rely on manually predefined evolutionary computation (EC) heuristic-optimizers and single-task training schemes, which may constrain the exploration of diverse heuristic algorithms and hinder the generalization of the resulting heuristics. To address these issues, we propose Meta-Optimization of Heuristics (MoH), a novel framework that operates at the optimizer level, discovering effective heuristic-optimizers through the principle of meta-learning. Specifically, MoH leverages LLMs to iteratively refine a meta-optimizer that autonomously constructs diverse heuristic-optimizers through (self-)invocation, thereby eliminating the reliance on a predefined EC heuristic-optimizer. These constructed heuristic-optimizers subsequently evolve heuristics for downstream tasks, enabling broader heuristic exploration. Moreover, MoH employs a multi-task training scheme to promote its generalization capability. Experiments on classic COPs demonstrate that MoH constructs an effective and interpretable meta-optimizer, achieving state-of-the-art performance across various downstream tasks, particularly in cross-size settings. Our code is available at: https://github.com/yiding-s/MoH.

## 1 Introduction

Heuristics have long been integral to solving combinatorial optimization problems (COPs), offering practical and efficient approaches when exact methods become computationally intractable due to their exponential time complexity. Over the past few decades, substantial progress has been achieved in human-designed heuristics. Notable examples include the Lin-Kernighan Heuristic (LKH) (Lin & Kernighan, 1973) for the Traveling Salesman Problem (TSP) and the Best Fit heuristic (Johnson et al., 1974) for the Bin Packing Problem (BPP). However, developing effective heuristics for COPs typically requires an in-depth understanding of each problem's unique structure and the specialized expertise to craft suitable heuristic strategies. As a result, the traditional approach to heuristic design is both time-intensive and significantly dependent on expert knowledge. This underscores the growing demand for more powerful approaches to accelerate the development of effective heuristics for COPs.

With the explosive advancements in large language models (LLMs) in recent years, the landscape of heuristic design has undergone a transformative shift (Jiang et al., 2024a; Liu et al., 2024b). A prominent trend involves leveraging LLMs to generate effective heuristics aimed at solving NP-hard COPs. Specifically, these methods typically utilize in-context learning to prompt LLMs to produce heuristics, which subsequently become integral components of (meta-)heuristic or learning-based solvers. Romera-Paredes et al. (2024) first demonstrated the feasibility of applying LLMs to heuristic design in this domain. Building on this foundational work, recent approaches have increasingly integrated LLMs with evolutionary computation (EC), giving rise to LLM-EC frameworks (Liu et al., 2024a; Ye et al., 2024; Dat et al., 2024; Zheng et al., 2025). These methods enhance heuristic design by using LLMs to carry out evolutionary operations like crossover and mutation to evolve heuristics.

---

[*]Corresponding authors.

Despite achieving promising results, existing LLM-EC approaches face two limitations. First, their search space is constrained by manually designed, predefined EC heuristic-optimizers (e.g., a fixed workflow of crossover followed by mutation), which may restrict the exploration of diverse heuristics and ultimately hinder the discovery of more powerful heuristics (Dat et al., 2024). Second, their optimization process is only designed for a single task (i.e., a fixed-size COP), which may limit the generalization of the evolved heuristics. Fig. 1 illustrates the generalization performance of EoH (Liu et al., 2024a), a representative LLM-EC approach, in optimizing improvement heuristics for TSP under various training settings. The results indicate a significant generalization challenge, as performance gaps widen with increasing problem size. Although incorporating cross-size datasets during training can partially mitigate this issue, the overall performance remains suboptimal on large problem sizes (i.e., different tasks).

To overcome these inherent limitations, we introduce Meta-Optimization of Heuristics (MoH), a novel framework leveraging the in-context reasoning and refinement capabilities of LLMs (Huang et al., 2022; Zelikman et al., 2024) to automate optimizer design. In this paper, we categorize optimizers into heuristic-optimizers and meta-optimizers based on their respective roles. *Heuristic-optimizers* are algorithms, such as traditional EC frameworks, that are used to generate or refine *heuristics* for COPs to improve solution quality, whereas *meta-optimizers* are higher-level procedures that adapt and enhance these heuristic-optimizers. Technically, MoH implements an iterative meta-optimization module within a multi-task framework to encourage both exploration and generalization.

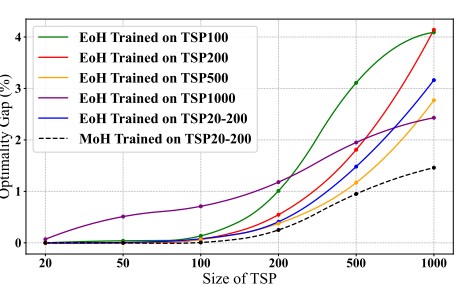

Figure 1: Generalization performance of the evolved improvement heuristics for TSP.

At each iteration, the meta-optimizer generates a diverse population of candidate heuristic-optimizers through (self-)invocation. The most promising heuristic-optimizer, evaluated by its effectiveness on downstream tasks in optimizing task-specific heuristics, is selected to become the meta-optimizer in the subsequent iteration. By doing so, the heuristic-optimizers are improved to generate more effective heuristics (see Fig. 2). With its innovative meta-optimization, MoH extends beyond traditional fixed EC optimization frameworks, facilitating broader exploration of the heuristic search space and operating at a higher abstraction level than existing approaches.

Our contributions are summarized as follows: 1) We propose MoH, a novel framework that highlights meta-optimization for producing effective COP heuristics. MoH enables broader heuristic exploration by autonomously discovering effective optimization strategies through an iterative meta-optimization module, thereby addressing inherent limitations of existing LLM-EC approaches. 2) We position MoH within a multi-task training framework to enhance its generalization capability to unseen tasks. 3) Extensive experiments across multiple heuristic algorithms and classical COPs demonstrate that MoH is able to generate effective and interpretable meta-optimizers that consistently outperform baselines. Notably, the resulting heuristics exhibit strong performance on large COP instances.

## 2 PRELIMINARIES

In this section, we first introduce three canonical COPs, TSP, BPP and CVRP, followed by an introduction to existing LLM-EC approaches and a high-level comparison with our proposed MoH.

**Traveling Salesman Problem.** Traveling Salesman Problem (TSP) is a well-known COP (Applegate, 2006). A TSP instance is defined over a complete graph $\mathcal{G} = \{\mathcal{V}, \mathcal{E}\}$, where $\mathcal{V} = \{v_1, \ldots, v_n\}$ is the set of cities and $\mathcal{E} = \{e(v_i, v_j)|v_i, v_j \in \mathcal{V}, i \neq j\}$ is the set of edges, representing possible travel routes between cities. Each edge $e(v_i, v_j)$ is associated with a distance $d_{ij}$, where $d : \mathcal{V} \times \mathcal{V} \to \mathbb{R}^+$ defines the travel cost between any pair of cities. The objective is to find a Hamiltonian cycle (i.e., a permutation of $\mathcal{V}$ that starts and ends at the same city) with the minimum total travel cost, subject to the constraint that each city is visited exactly once before returning to the starting city.

**Bin Packing Problem.** The Bin Packing Problem (BPP) aims to pack a set of items $\{i_1, i_2, \ldots, i_n\}$, each with an associated weight $w_i$, into a minimum number of bins of fixed capacity $C$, such that the total weight of items assigned to each bin does not exceed $C$. BPP can be categorized into online and

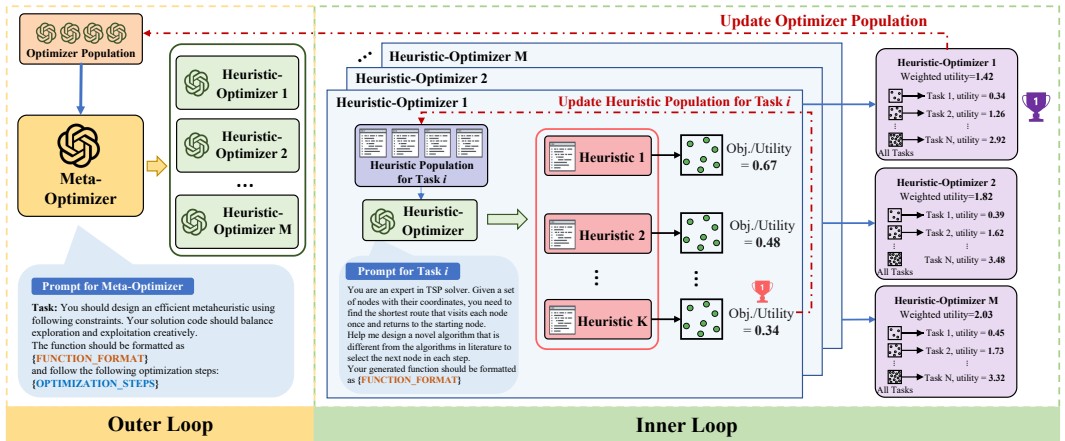

Figure 2: Overview of MoH. In iteration $t$, the current meta-optimizer $\mathcal{I}_{t-1}^*$ generates $M$ candidate heuristic-optimizers in the *outer loop*. Each candidate heuristic-optimizer is then evaluated through the *inner loop*, where it generates $K$ heuristics that are applied to $N$ downstream tasks. For each task, the best heuristic is selected, and its utility contributes to the overall utility of the heuristic-optimizer. After aggregating utility scores across all tasks, the heuristic-optimizer with the highest utility is selected as the new meta-optimizer $\mathcal{I}_t^*$.

offline variants depending on the availability of item information. In the online BPP (Seiden, 2002), items arrive sequentially in an unknown order. For each arriving item, a placement decision must be made immediately and irrevocably without knowledge of future items. The objective remains to minimize the total number of bins used while respecting capacity constraints. In contrast, the offline BPP (Coffman Jr et al., 1984) assumes complete prior knowledge of all item sizes and weights, enabling global optimization over the entire item set. This setting allows algorithms to exploit full information to search for higher-quality packing configurations.

**Capacitated Vehicle Routing Problem.** The Capacitated Vehicle Routing Problem (CVRP) is also a classical NP-hard combinatorial optimization problem widely studied in the fields of logistics and operations research (Toth & Vigo, 2002). It builds upon the classic TSP by incorporating the crucial real-world constraint of limited vehicle capacity. Specifically, a CVRP instance can be defined over a complete graph $\mathcal{G} = \{\mathcal{V} \cup v_0, \mathcal{E}\}$, where $\mathcal{V} = \{v_1, \ldots, v_n\}$ denotes the set of customer nodes, $v_0$ represents the depot, and $\mathcal{E} = \{e(v_i, v_j) | v_i, v_j \in \mathcal{V} \cup v_0, i \neq j\}$ is the edge set that includes all the possible travel routes between any two nodes, either customers or depot. Each edge $e(v_i, v_j)$ is associated with a non-negative travel cost or distance $d_{ij}$, where $d : (\mathcal{V} \cup v_0) \times (\mathcal{V} \cup v_0) \to \mathbb{R}^+$ defines the travel cost between any pair of two nodes. Each customer $v_i \in \mathcal{V}$ has a demand $q_i > 0$, while the depot has $q_0 = 0$. The fleet consists of $m$ vehicles, each with capacity $Q$. The objective is to determine a set of $m'$ routes (usually $m' < m$) with minimized total travel cost across all routes, while satisfying the following constraints: 1) each route starts and ends at the depot, 2) each customer is visited exactly once by a single vehicle, and 3) the total demand on any route does not exceed $Q$.

**Existing LLM-EC Approaches.** In the domain of LLM-driven automatic design and generation of executable heuristics, early approaches (Liu et al., 2024a) leverage a fixed EC heuristic-optimizer to discover effective heuristics through LLMs for solving COPs. Specifically, they maintain a fixed-size population of heuristics tailored to a specific COP task. The EC heuristic-optimizer refines this population by iteratively selecting promising candidates and applying crossover and mutation operations to generate improved heuristic variants. While this method can quickly converge to reasonably good heuristics, its performance may be constrained by the insufficient exploration of the vast search space. Subsequent works propose various EC variants, such as incorporating a reflection mechanism (Ye et al., 2024) or integrating with Monte Carlo Tree Search (MCTS) (Zheng et al., 2025), however, these methods still suffer from limited exploration or high computational cost. In summary, existing approaches primarily focus on heuristic design using a fixed EC heuristic-optimizer, whereas MoH targets optimizer design, going a step further by enabling automatic design of such high-level optimization frameworks themselves. This allows for more flexible and potentially more effective heuristic generation for downstream tasks, as illustrated in Fig. 2. We believe our approach

offers fresh insights into the field by introducing a conceptually and methodologically meaningful advancement in the use of LLMs for combinatorial optimization, addressing both optimization framework discovery and heuristic evolution in a unified and scalable manner.

## 3 METHODOLOGY

An overview of MoH is shown in Fig. 2, which features a two-level optimization process: *an outer loop for optimizer design* and *an inner loop for heuristic design.* Inspired by recent advances in LLMs (Zhou et al., 2022; Zelikman et al., 2024), MoH aims to construct a meta-optimizer capable of generating effective optimization strategies and improving heuristics on downstream tasks concurrently. Concretely, in the outer loop, the meta-optimizer produces a diverse population of candidate heuristic-optimizers. Then, each generated heuristic-optimizer is leveraged in the inner loop to evolve task-specific heuristics for downstream tasks. After evaluating the heuristics on the validation dataset, the candidate heuristic-optimizer with the highest utility score is selected as the new meta-optimizer for the next iteration, enabling MoH to iteratively discover increasingly effective optimization strategies. Moreover, MoH is inherently suited for a multi-task training setting by maintaining diversity among tasks, thereby enhancing its ability to explore a broader range of heuristics, leading to improved performance across diverse tasks. Examples of seed (or initial) and generated meta-optimizers are provided in Appendix E. In the following sections, we present the technical details of our proposed MoH framework.

### 3.1 PROBLEM FORMULATION

Suppose there are $N$ downstream tasks, each corresponding to a heuristic design task for a COP. For each task $i$, let $h_i^{\mathcal{I}}$ denote the heuristic found by the heuristic-optimizer $\mathcal{I}$, $\mathcal{D}_i$ the validation dataset, and $U_i(h_i^{\mathcal{I}}, \mathcal{D}_i)$ the heuristic utility function evaluating the performance of $h_i^{\mathcal{I}}$ on $\mathcal{D}_i$. The objective of *heuristic design* is to discover the best heuristic $\tilde{h}_i^{\mathcal{I}}$ using heuristic-optimizer $\mathcal{I}$:

$$\tilde{h}_i^{\mathcal{I}} = \arg \max_{h_i^{\mathcal{I}} \in \mathbb{H}_i} U_i(h_i^{\mathcal{I}}, \mathcal{D}_i), \tag{1}$$

where $\mathbb{H}_i$ denotes the heuristic search space, comprising all possible heuristics for the task $i$. The heuristic utility function $U_i(h_i^{\mathcal{I}}, \mathcal{D}_i)$ is defined as the negative of the solution optimality gap. Most studies so far integrate EC as the heuristic-optimizer $\mathcal{I}$ within LLMs to perform heuristic design. While these LLM-EC approaches offer a certain degree of flexibility, they struggle to effectively explore the vast heuristic search space due to the rigid structure of the fixed heuristic-optimizer $\mathcal{I}$. Additionally, their heuristic design process necessitates separate training for each task $i$, making it computationally expensive. An alternative is to incorporate diverse instances from $N$ tasks into the training dataset. However, this simple mixture of data results in suboptimal performance, as a single COP heuristic usually struggles to adapt effectively across different tasks, consistent with the No Free Lunch Theorem (Wolpert & Macready, 1997).

To address the limitations, MoH directly searches for optimizers rather than relying on a fixed one, i.e., the optimizer design process. The objective of optimizer design is formally defined as:

$$\mathcal{I}^* \leftarrow \tilde{\mathcal{I}} = \arg \max_{\mathcal{I}} \sum_{i=1}^{N} w_i \cdot U_i(\tilde{h}_i^{\mathcal{I}}, \mathcal{D}_i), \text{ with } \tilde{h}_i^{\mathcal{I}} = \arg \max_{h_i^{\mathcal{I}} \in \mathbb{H}_i} U_i(h_i^{\mathcal{I}}, \mathcal{D}_i), \tag{2}$$

where $\mathcal{I}^*$ is the meta-optimizer, and $w_i$ is the task weight defined as $w_i = \frac{s_i}{\sum_{j=1}^{N} s_j}$, with $s_i$ representing the problem size of the $i$-th downstream task. In essence, MoH extends Eq. (1) by introducing an outer loop for meta-optimization. In this outer loop, the meta-optimizer $\mathcal{I}^*$ produces a population of candidate heuristic-optimizers through (self-)invocation. The best heuristic-optimizer $\tilde{\mathcal{I}}$, as evaluated by the optimizer utility function $\mathbf{U}(\tilde{\mathcal{I}}) = \sum_{i=1}^{N} w_i \cdot U_i(\tilde{h}_i^{\tilde{\mathcal{I}}}, \mathcal{D}_i)$, is then selected to serve as the new meta-optimizer $\mathcal{I}^*$ in the next iteration.

### 3.2 OVERALL WORKFLOW

We summarize the MoH training workflow in Alg. 1 and detail each step as follows. We initialize each downstream heuristic design task $i$ with a heuristic population $\mathcal{H}_i$. This is achieved by prompting

---

**Algorithm 1:** MoH Training Workflow

---

**Input:** Number of downstream tasks $N$, Number of iterations $T$, Seed meta-optimizer $\mathcal{I}_0$;
**Output:** Meta-optimizer $\mathcal{I}_T^*$, Heuristic populations $\mathcal{H}$ across all tasks;
**Function U** $(\mathcal{I})$:

  $u \leftarrow 0$
  **for** $i = 1, \ldots, N$ **do**

   $\tilde{h}_i^{\mathcal{I}} \leftarrow \mathcal{I}(\mathcal{H}_i, U_i(\cdot), \text{LLM}, \text{Prompt}, \text{"Task i"})$:

   ┌─────────────────────────────────────────────────────────────┐
   **Steps for heuristic design with heuristic-optimizer $\mathcal{I}$:**
   a. Generate a group of heuristics using $\mathcal{I} \rightarrow \{h_{i,1}^{\mathcal{I}}, \ldots, h_{i,K}^{\mathcal{I}}\}$
   b. Evaluate the utility score of each heuristic through $U_i(h_{i,k}^{\mathcal{I}}, \mathcal{D}_i)$, $\forall k \in [1, K]$
   c. Update $\mathcal{H}_i = \text{Top}\mathcal{K}(\mathcal{H}_i \cup \{h_{i,1}^{\mathcal{I}}, \ldots, h_{i,K}^{\mathcal{I}}\})$ by utility and return the best heuristic $\tilde{h}_i^{\mathcal{I}}$
   └─────────────────────────────────────────────────────────────┘

   $u \leftarrow u + \omega_i \cdot U_i(\tilde{h}_i^{\mathcal{I}}, \mathcal{D}_i)$
  **return** $u$

▶▶ Initialize heuristic and optimizer populations $\mathcal{H} = \{\mathcal{H}_1, \ldots, \mathcal{H}_N\}, \mathcal{P} = \{\mathcal{I}_0\}$
**for** $t = 1, \ldots, T$ **do**

  $\tilde{\mathcal{I}}_t \leftarrow \mathcal{I}_{t-1}^*(\mathcal{P}, \mathbf{U}(\cdot), \text{LLM}, \text{Prompt}, \text{"Optimizer"})$:

   ┌─────────────────────────────────────────────────────────────┐
   **Steps for optimizer design with meta-optimizer $\mathcal{I}_{t-1}^*$:**
   a. Generate a group of heuristic-optimizers via (self-)invocation of $\mathcal{I}_{t-1}^* \rightarrow \{\mathcal{I}_t^1, \ldots, \mathcal{I}_t^M\}$
   b. Evaluate the utility score of each heuristic-optimizer through $\mathbf{U}(\mathcal{I}_t^j)$, $\forall j \in [1, M]$
   c. Update $\mathcal{P} = \text{Top}\mathcal{K}(\mathcal{P} \cup \{\mathcal{I}_t^1, \ldots, \mathcal{I}_t^M\})$ by utility and return the best heuristic-optimizer $\tilde{\mathcal{I}}_t$
   └─────────────────────────────────────────────────────────────┘

  $\mathcal{I}_t^* \leftarrow \tilde{\mathcal{I}}_t$
**return** $\mathcal{I}_T^*, \mathcal{H}$

---

LLMs to generate diverse heuristic ideas in natural language, accompanied by their corresponding code implementations. We also initialize an optimizer population $\mathcal{P}$ using a given seed meta-optimizer (see Fig. 8), which serves as the starting point and reference baseline for subsequent iterations. Concretely, at iteration $t$, the current meta-optimizer $\mathcal{I}_{t-1}^*$ is used to generate a set of candidate heuristic-optimizers $\{\mathcal{I}_t^1, \ldots, \mathcal{I}_t^M\}$. This generation is accomplished via (self-)invocation of $\mathcal{I}_{t-1}^*$ using LLMs, with prompts constructed from the information in the optimizer population $\mathcal{P}$. Then, each candidate heuristic-optimizer $\mathcal{I}_t^j$ employs its own LLM-generated optimization strategy to evolve heuristics across all downstream tasks, resulting in updated heuristic populations for all $N$ downstream tasks $\{\{h_{i,1}^{\mathcal{I}_t^j}, \ldots, h_{i,K}^{\mathcal{I}_t^j}\}\}_{i=1}^N$. After evaluation, the best heuristic for each task is collected, yielding $\{\tilde{h}_1^{\mathcal{I}_t^j}, \ldots, \tilde{h}_N^{\mathcal{I}_t^j}\}$. The utility score of each candidate heuristic-optimizer is thus calculated as $\mathbf{U}(\mathcal{I}_t^j) = \sum_{i=1}^N w_i \cdot U_i(\tilde{h}_i^{\mathcal{I}_t^j}, D_i)$. The candidate heuristic-optimizer with the highest utility score is selected as the meta-optimizer $\mathcal{I}_t^*$ for the next iteration. More details on population management can be found in Section 3.3. During inference, the meta-optimizer $\mathcal{I}_T^*$ can be deployed on new tasks that differ from those encountered during training, such as tasks with larger problem sizes. By performing several rounds of heuristic design using $\mathcal{I}_T^*$, MoH can yield an effective heuristic tailored to this task.

## 3.3 DETAILED IMPLEMENTATION

As key subroutines of MoH, we further elaborate on the optimizer and heuristic generation (i.e., step (a) in the heuristic and optimizer design processes in Alg. 1). The main entities are as follows.

**Individual and Population Structure.** In the MoH framework, an individual is defined as a structured entity comprising three components: 1) a code implementation (String), 2) a high-level natural language description of the core strategy (String), and 3) a utility score reflecting its performance (Float). This unified individual format is adopted for both heuristics and optimizers. To ensure diversity and stability, we preserve a population with 10 individuals for both heuristic populations $\mathcal{H}$ and optimizer population $\mathcal{P}$.

---

**Optimizer Signature**

```python
def optimize_algorithm(
    population: dict,
    utility: callable[[dict], float],
    language_model: class 'LanguageModel'
    subtask_prompt: str,
    subtask: str
) -> Tuple[str, str, float]:
```

---

Figure 3: This signature applies to meta-optimizers and heuristic-optimizers. Its detailed implementation is generated by LLMs, enabling recursive or iterative refinement of optimization strategies.

**Population Management.** Individuals in the population are ranked by their utility scores. When a new candidate arrives, it is compared to the current worst-performing individual. If the candidate's utility score is higher, it replaces that individual. After each insertion, the population is re-sorted to maintain the utility-based ranking. This structure is efficiently managed using a heap.

**Optimizer Signature.** As shown in Fig. 3, the optimizer is formatted as a callable function that takes the following inputs: 1) *population:* the population structure defined above, 2) *utility:* a utility function that evaluates the performance of an individual and returns its utility score, 3) *language_model:* an LLM for generating heuristics or heuristic-optimizers, 4) *subtask_prompt:* a task-specific prompt to guide the optimization, and 5) *subtask:* a string specifying the name of the task. The optimizer function returns the best individual discovered during the optimization process. Notably, it is designed to support recursive invocation, allowing it to take its own implementation as input through the population parameter in the first outer loop iteration.

**Optimizer Generation Procedure.** The heuristic-optimizer generation procedure in the outer loop iteration $t$ follows the standardized steps below: 1) *Individual Selection:* The current meta-optimizer $\mathcal{I}_{t-1}^*$ uses its LLM-generated strategy to select promising candidate heuristic-optimizers from the optimizer population $\mathcal{P}$. This step aims to balance the exploitation of high-utility individuals with the exploration of diverse candidates. 2) *Idea Generation:* The iterative improvement of algorithms generated by LLMs critically depends on algorithmic reasoning articulated in natural language, as evidenced by (Wang et al., 2024). Consequently, the meta-optimizer is encouraged to prompt LLMs to propose exploratory or refinement ideas based on the heuristic-optimizers selected in the first step. 3) *Implementation Generation:* Guided by the generated ideas and task-specific prompts, the LLM refines or generates new code implementations of selected optimizers through (self-)invocation of $\mathcal{I}_{t-1}^*$, producing a set of candidate heuristic-optimizers $\{\mathcal{I}_t^1, \ldots, \mathcal{I}_t^M\}$. Each candidate is evaluated using the optimizer utility function $\mathbf{U}(\cdot)$, and the best-performing heuristic-optimizer is selected as the new meta-optimizer $\mathcal{I}_t^*$ for the next iteration. Note that our optimizer structure enables flexible exploration of various optimization strategies. While the specific behavior of a generated heuristic-optimizer may vary depending on prior heuristic-optimizers, prompts, and LLM versions, their procedures generally adhere to the above three steps. Appendix E presents examples of LLM-generated meta-optimizers, some of which resemble traditional metaheuristics, while others exhibit hybrid or unconventional strategies.

**Heuristic Generation Procedure.** Given a heuristic-optimizer $\mathcal{I}_t^j$ generated by the meta-optimizer $\mathcal{I}_{t-1}^*$ in outer loop iteration $t$, the inner loop heuristic generation procedure for downstream task $i$ follows the standardized steps below: 1) *Individual Selection:* The heuristic-optimizer $\mathcal{I}_t^j$ employs its LLM-generated strategy to select promising candidate heuristics from $\mathcal{H}_i$ for evolution. 2) *Idea Generation:* The heuristic-optimizer $\mathcal{I}_t^j$ prompts LLMs to propose exploratory or refinement ideas based on the heuristics selected in the first step. 3) *Implementation Generation:* Guided by the generated ideas and task-specific prompts, the LLM generates or refines heuristic implementations, resulting in a set of candidate heuristics $\{h_{i,1}^{\mathcal{I}_t^j}, \ldots, h_{i,K}^{\mathcal{I}_t^j}\}$. Each candidate heuristic is evaluated using its corresponding heuristic utility function $U_i(\cdot)$, and the heuristic population $\mathcal{H}_i$ is updated thereafter.

Despite their procedural similarities, the key differences between optimizer generation in the outer loop and heuristic generation in the inner loop are as follows: 1) *Optimization Target:* The outer loop focuses on generating heuristic-optimizers using the meta-optimizer, whereas the inner loop

Table 1: Results of constructive and improvement heuristics on TSP.

| Methods | Train | | | | | | | | Generalization | | | | Average Gap ↓ |
|---|---|---|---|---|---|---|---|---|---|---|---|---|---|
| | 20 | | 50 | | 100 | | 200 | | 500 | | 1000 | | |
| | Obj. ↓ | Gap ↓ | Obj. ↓ | Gap ↓ | Obj. ↓ | Gap ↓ | Obj. ↓ | Gap ↓ | Obj. ↓ | Gap ↓ | Obj. ↓ | Gap ↓ | |
| Concorde | 3.840 | - | 5.715 | - | 7.766 | - | 10.679 | - | 16.519 | - | 23.104 | - | - |
| OR-Tools | 3.840 | 0.000% | 5.715 | 0.001% | 7.772 | 0.089% | 10.944 | 2.478% | 17.259 | 4.479% | 24.262 | 5.011% | 2.010% |
| Nearest Neighbor | 4.602 | 19.806% | 7.055 | 23.406% | 9.636 | 24.072% | 13.374 | 25.228% | 20.691 | 25.252% | 28.990 | 25.474% | 23.873% |
| *Constructive Heuristic* | | | | | | | | | | | | | |
| Funsearch | 4.261 | 11.000% | 6.523 | 14.162% | 9.018 | 16.109% | 12.615 | 18.143% | 19.531 | 18.242% | 27.571 | 19.332% | 16.165% |
| EoH | 4.204 | 9.408% | 6.402 | 12.007% | 8.774 | 12.974% | 12.233 | 14.548% | 19.029 | 15.196% | 26.890 | 16.390% | 13.420% |
| ReEvo | 4.197 | 9.250% | 6.399 | 11.966% | 8.786 | 13.133% | 12.217 | 14.403% | 19.035 | 15.232% | 26.818 | 16.076% | 13.343% |
| HSEvo | 4.108 | 6.897% | **6.280** | **9.881%** | 8.705 | 12.102% | 12.208 | 14.320% | 19.550 | 18.349% | 27.431 | 18.727% | 13.379% |
| MCTS-AHD | 4.107 | 6.882% | 6.332 | 10.807% | 8.735 | 12.499% | 12.165 | 13.921% | 19.036 | 15.240% | 26.814 | 16.060% | 12.568% |
| MoH (Ours) | **4.104** | **6.837%** | 6.280 | 9.893% | **8.654** | **11.444%** | **12.100** | **13.307%** | **18.869** | **14.224%** | **26.581** | **15.049%** | **11.792%** |
| *Improvement Heuristic* | | | | | | | | | | | | | |
| EoH-GLS | 3.840 | 0.000% | 5.715 | 0.000% | 7.768 | 0.024% | 10.716 | 0.342% | 16.714 | 1.176% | 23.747 | 2.781% | 0.721% |
| HSEvo-GLS | 3.840 | 0.000% | 5.715 | 0.000% | 7.768 | 0.028% | 10.715 | 0.328% | 16.729 | 1.266% | 23.719 | 2.660% | 0.714% |
| ReEvo-GLS | 3.840 | 0.000% | 5.715 | 0.000% | 7.768 | 0.021% | 10.715 | 0.331% | 16.741 | 1.344% | 23.731 | 2.715% | 0.735% |
| MoH-GLS (Ours) | **3.840** | **0.000%** | **5.715** | **0.000%** | **7.767** | **0.012%** | **10.711** | **0.291%** | **16.674** | **0.936%** | **23.445** | **1.476%** | **0.453%** |
| ReEvo-KGLS | 3.840 | 0.000% | 5.715 | 0.000% | 7.766 | 0.003% | 10.704 | 0.221% | 16.681 | 0.976% | 23.473 | 1.595% | 0.466% |
| HSEvo-KGLS | 3.840 | 0.000% | 5.715 | 0.000% | 7.767 | 0.004% | 10.704 | 0.221% | 16.678 | 0.958% | 23.478 | 1.615% | 0.466% |
| MCTS-AHD-KGLS | 3.840 | 0.000% | 5.715 | 0.000% | 7.767 | 0.006% | 10.702 | 0.204% | 16.662 | 0.867% | 23.425 | 1.389% | 0.411% |
| MoH-KGLS (Ours) | **3.840** | **0.000%** | **5.715** | **0.000%** | **7.766** | **0.002%** | **10.699** | **0.177%** | **16.652** | **0.805%** | **23.419** | **1.363%** | **0.391%** |

applies each generated heuristic-optimizer to improve heuristics across all downstream tasks. 2) *Invocation Frequency:* In the outer loop, the meta-optimizer is invoked once per iteration to generate $M$ candidate heuristic-optimizers. In contrast, during the inner loop, each heuristic-optimizer is individually applied to generate $K$ candidate heuristics for each downstream task. Consequently, the invocation frequency in the inner loop is higher than in the outer loop. Detailed prompts for optimizer and heuristic generation can be found in Appendix D.

## 4 EXPERIMENTS

We conduct extensive experiments to optimize various heuristic algorithms on several classical COP benchmarks, including TSP, CVRP, online and offline BPP (see Table 1, 2 and 3), with a primary focus on cross-size generalization (see Appendix A for cross-distribution and cross-problem settings discussion). Additional results on other benchmarks (e.g., TSPLib) and other optimization problems are provided in Table 8, 9 and 17 (see Appendix C). All experiments are conducted on servers with NVIDIA GeForce RTX 4090 GPUs and AMD Ryzen Threadripper PRO 7975WX CPU @ 4GHz.

**Heuristic Settings.** 1) *Constructive Heuristic for TSP:* In constructive heuristics, a solution is built incrementally, starting from a random node and iteratively selecting the next promising node based on a predefined rule. The selected node is then appended to the current route to form a valid tour step by step. Since constructive heuristics focus on local optimization at each step rather than the global optimum, their performance is often suboptimal compared to other heuristic algorithms. 2) *Improvement Heuristic for TSP:* Guided Local Search (GLS) (Voudouris et al., 2010) is a metaheuristic that penalizes frequently used edges in local optima, steering the search away from less promising regions. Specifically, it modifies the cost landscape by adjusting the distance matrix, adding penalties to certain edges to prevent their repeated selection in subsequent iterations. In our experiment, we compare two GLS implementations from (Liu et al., 2024a) and (Ye et al., 2024). The implementation in (Liu et al., 2024a) follows a standard GLS approach, combining a basic local search method with dynamic edge penalties to guide the search. In contrast, the approach in (Ye et al., 2024) aligns more closely with Knowledge-Guided Local Search (KGLS) (Arnold & Sörensen, 2019), incorporating domain-specific knowledge from the distance matrix to enhance the standard GLS framework. In our experiments, we use GLS and KGLS to represent two different settings. 3) *Online BPP:* We follow the settings of (Romera-Paredes et al., 2024) to develop a heuristic that assigns incoming items to bins in real time. The heuristic utilizes a scoring function to determine the most suitable bin for each item dynamically (Angelopoulos et al., 2023). We evaluate the generated heuristics on 100 Weibull instances for each problem size, ranging from 1,000 to 10,000, with bin capacities varying from 100 to 500. The lower bound $lb$ for each instance is calculated as the ceiling of the total item weight divided by the capacity of a single bin: $lb = \left\lceil \frac{\sum_{i=1}^{n} w_i}{c} \right\rceil$, where $w_i$ is the weight of the

Table 2: Results on Online BPP.

| Bin Capacity | Item Size | Best Fit | First Fit | FunSearch | EoH | ReEvo | HSEvo | MCTS-AHD | MoH (Ours) |
|---|---|---|---|---|---|---|---|---|---|
| 100 | 1k | 4.621% | 5.038% | 3.165% | 3.294% | 3.475% | 3.748% | **2.543%** | 2.553% |
|  | 5k | 4.149% | 4.488% | 2.165% | 0.827% | 2.022% | 1.088% | 1.769% | **0.600%** |
|  | 10k | 4.030% | 4.308% | 2.008% | 0.436% | 1.821% | 0.734% | 1.647% | **0.414%** |
| 200 | 1k | 1.825% | 2.025% | 0.938% | 1.645% | 1.825% | 1.825% | 1.238% | **0.848%** |
|  | 5k | 1.555% | 1.665% | 0.543% | 0.366% | 1.549% | 1.555% | 1.062% | **0.262%** |
|  | 10k | 1.489% | 1.578% | 0.459% | 0.188% | 1.489% | 1.489% | 1.036% | **0.141%** |
| 300 | 1k | 1.131% | 1.265% | 0.654% | 1.086% | 1.131% | 1.131% | 0.922% | **0.581%** |
|  | 5k | 0.919% | 0.984% | 0.352% | 0.254% | 0.919% | 0.919% | 0.785% | **0.161%** |
|  | 10k | 0.882% | 0.924% | 0.316% | 0.115% | 0.882% | 0.882% | 0.765% | **0.079%** |
| 400 | 1k | 0.815% | 0.835% | 0.519% | 0.815% | 0.815% | 0.815% | 0.755% | **0.498%** |
|  | 5k | 0.624% | 0.672% | 0.275% | 0.191% | 0.621% | 0.624% | 0.608% | **0.104%** |
|  | 10k | 0.603% | 0.639% | 0.243% | 0.098% | 0.595% | 0.603% | 0.579% | **0.054%** |
| 500 | 1k | 0.546% | 0.522% | **0.324%** | 0.695% | 0.546% | 0.546% | 0.496% | 0.373% |
|  | 5k | 0.472% | 0.507% | 0.214% | 0.119% | 0.472% | 0.472% | 0.447% | **0.090%** |
|  | 10k | 0.448% | 0.487% | 0.196% | 0.075% | 0.445% | 0.448% | 0.430% | **0.032%** |
| Average | | 1.607% | 1.729% | 0.825% | 0.680% | 1.240% | 1.125% | 1.006% | **0.453%** |

item $i$ and $c$ is the bin capacity (Martello & Toth, 1990). 4) *CVRP and Offline BPP:* For CVRP and Offline BPP, we develop heuristics under the Ant Colony Optimization (ACO) framework (Ye et al., 2024; 2023), a setting also adopted by (Dat et al., 2024) and (Zheng et al., 2025). Specifically, ACO maintains a pheromone matrix $\tau \in \mathbb{R}^{n \times n}$ and a heuristic matrix $\eta \in \mathbb{R}^{n \times n}$, where $n$ represents the number of vertices in each problem. Each entry $\tau_{i,j}$ represents the priority of edge $(i, j)$ selection in solution construction, which is iteratively updated based on previous solutions. For the heuristic information in $\eta_{i,j}$, it encodes the problem-specific guidance for choosing edge $(i, j)$, reflecting its local contribution to final solution quality. Our objective is to design more effective heuristic matrices $\eta$ by leveraging problem-specific input features.

**Baselines.** 1) *Traditional methods:* We employ Concorde (Applegate et al., 2003) and OR-Tools (Furnon & Perron, 2023) to solve TSP, and compare with classic heuristics, including Nearest Neighbor for TSP, Best Fit and First Fit for online BPP. For OR-Tools, we use guided local search as the local search strategy. The time limit for solving each TSP instance is set to 20s for problem size $\leq 100$ and 40s for problem size $\geq 200$. 2) *LLM-based methods:* We compare MoH with five representative approaches: FunSearch (Romera-Paredes et al., 2024), EoH (Liu et al., 2024a), ReEvo (Ye et al., 2024), HSEvo (Dat et al., 2024), and MCTS-AHD (Zheng et al., 2025). We rerun their publicly available implementations in our training settings, as detailed below. 3) *Neural methods:* We also benchmark against neural solvers, such as POMO (Kwon et al., 2020), LEHD (Luo et al., 2023) and SIL (Luo et al., 2024), with results reported in Appendix C.

**Training and Inference.** To ensure a fair comparison, all LLM-based methods are trained under identical experimental conditions for each problem setting and evolved without relying on any predefined seed heuristic. In the TSP scenario, all methods are trained on cross-size datasets comprising four tasks: TSP20, 50, 100 and 200, and generalized to larger instances of sizes 500 and 1000. In the online BPP scenario, all methods are trained on two tasks: 1,000 items with bin capacity 1,000, and 5,000 items with bin capacity 1,000. During training, we fix the number of outer loop iterations to $T = 10$ and maintain a population size of 10 for both heuristic-optimizer and heuristic populations. We control the computational budget of each method by limiting the number of heuristic evaluations to 1,000. A detailed analysis of computational costs is provided in Table 10, 11 and 12 (see Appendix C). At inference stage, the trained meta-optimizer is executed for 10 iterations. For the final results of heuristic performance, we use 128 instances for TSP heuristic evaluation and 100 instances for online BPP heuristic evaluation. All results reported in Tables 1 and 2 reflect the average performance over the test dataset of the best-performing heuristic identified across three independent runs. All experiments use GPT 4o-mini (2024-07-18) as base LLM.

## 4.1 EMPIRICAL RESULT

Table 1 presents a comprehensive comparison of our proposed MoH against baselines on TSP. The table includes best results for both constructive and improvement heuristics across TSP20-1000

Table 3: Results on CVRP and Offline BPP.

| Problems | CVRP | | | | Offline BPP | | | |
|---|---|---|---|---|---|---|---|---|
| | Train | | Generalization | | Train | | Generalization | |
| Methods | 20 | 50 | 100 | 200 | N=100, C=150 | N=500, C=150 | N=500, C=300 | N=1000, C=300 |
| ReEvo | 4.826 | 9.339 | 15.901 | 28.224 | 41.984 | 207.406 | 102.438 | 204.438 |
| HSEvo | 4.858 | 9.250 | 15.940 | 28.598 | 41.766 | 205.500 | 102.438 | 204.656 |
| MCTS-AHD | 4.843 | 9.165 | 15.630 | 28.041 | 41.656 | **204.609** | 102.625 | 204.734 |
| MoH | **4.704** | **9.059** | **15.563** | **27.512** | **41.625** | 205.453 | **102.125** | **203.750** |

instances in three independent runs. The optimality gap is calculated as the difference in cost between each heuristic's solution and the optimal solution, obtained using the Concorde solver (Applegate et al., 2003). In the constructive heuristic setting, MoH achieves the lowest average optimality gap of 11.792%, significantly outperforming existing LLM-based approaches. In the improvement heuristic setting, we evaluate MoH using both GLS and KGLS variants. Our approach consistently achieves the lowest optimality gap of 0.391%, demonstrating superior solution quality across various TSP instances. Moreover, MoH demonstrates strong generalization performance on large-scale instances across both settings. These results confirm the effectiveness and adaptability of our approach in both heuristic categories and across different data regimes. Additional heuristic performance results with different baselines on TSPLib (Reinelt, 1991) are presented in Table 8 and 9 (see Appendix C). Detailed results for statistical performance are listed in Table 16.

Table 2 summarizes the best performance of MoH on the Online BPP across three independent runs, evaluated over a variety of settings with different bin capacities (100 to 500) and item set sizes (1k, 5k, 10k). The reported metric is the proportion of excess bins used relative to the theoretical lower bound. Our method outperforms all competing baselines and traditional heuristics Best Fit, First Fit across nearly all instance settings, achieving lower average bin usage. These results indicate that MoH also performs well on online packing tasks, demonstrating strong adaptability and generalization in dynamic environments. Appendix E provides examples of the best-performing heuristics discovered by MoH in large-scale settings across these problems. We further conducted experiments on additional COPs including CVRP and Offline BPP, with the best objective values across three runs for different problem sizes summarized in Table 3. Results indicate that MoH achieves optimal or near-optimal solution costs in almost all settings across different problem scales.

## 4.2 ABLATION STUDY AND FURTHER ANALYSIS

In this section, we provide a more in-depth analysis of MoH. The experiments shown in Fig. 4 and Tables 4 and 5 are conducted under the improvement heuristic (i.e., GLS) setting on TSP. Results are averaged over three runs, using TSP100 and TSP200 as the downstream tasks during training.

**Idea Generation.** A key strength of LLMs lies in their powerful natural language processing capabilities. Integrating code generation and optimization tasks with natural language algorithm descriptions is therefore a natural approach. We incorporate these descriptions into the code generation process as *ideas*, enabling LLMs to fully utilize their language understanding abilities. This strategy goes beyond simple repeated sampling within the code space, allowing LLMs to explore a broader and more diverse solution space. As shown in Fig. 4 and Table 5, we compare the performance of methods with and without natural language ideas. The utility score reflects the average performance across two downstream tasks during training. The results clearly demonstrate that incorporating such natural language descriptions improves training performance.

**Different LLMs.** In Table 4 and Table 14, we evaluate several LLMs beyond GPT 4o-mini (2024-07-18) to assess the adaptability of MoH, including o1-mini (2024-09-12), deepseek-v3 (2024/12/26) and Qwen-plus-0919. The results demonstrate that our framework performs well across different LLMs. Furthermore, we observe that LLMs with more learnable parameters and larger context windows tend to produce longer and more complex optimization strategies. However, increased heuristic-optimizer complexity does not necessarily lead to better performance of downstream heuristics. For instance, the more advanced o1-mini does not outperform 4o-mini on large-scale TSP instances.

Table 4: Ablation results of MoH-GLS on different LLMs for TSP.

| LLM | 100 | 200 | 500 | 1000 |
|---|---|---|---|---|
| 4o mini | 0.035% | **0.332%** | **1.045%** | **1.710%** |
| o1-mini | **0.024%** | 0.353% | 1.280% | 2.119% |
| Deepseek-V3 | 0.108% | 0.413% | 1.314% | 2.527% |
| Qwen-Plus | 0.057% | 0.375% | 1.753% | 3.277% |

Table 5: Ablation results of MoH-GLS on idea generation and population size for TSP.

| Setting | Population Size | w. idea | 100 | 200 | 500 | 1000 |
|---|---|---|---|---|---|---|
| | 1 | ✓ | 0.120% | 0.563% | 2.355% | 3.300% |
| | 5 | ✓ | 0.058% | 0.337% | 1.475% | 2.338% |
| | 10 | ✗ | 0.043% | 0.390% | 1.644% | 2.420% |
| Default | 10 | ✓ | **0.035%** | **0.332%** | **1.045%** | **1.710%** |

**Population size.** For each downstream task, MoH maintains a heuristic population that allows the heuristic-optimizer to iteratively select, reference and refine promising candidates. Given the well-established effectiveness of few-shot prompting, it is crucial to retain elite heuristics from previous iterations to guide subsequent optimization steps, supporting both exploration and exploitation. As shown in Fig. 4 and Table 5, we evaluate the impact of different population sizes during MoH training. The results indicate that a small population size limits the LLM's ability to effectively leverage top-performing candidates when generating improved ones. To balance computational cost, training time, and exploration breadth in downstream heuristic tasks, we set the population size to 10. This choice ensures sufficient solution diversity while keeping overhead manageable.

**Analysis of Meta-Optimizer.** We take a deeper look into the meta-optimizers generated by MoH. In Appendix E, we present representative examples and analyze their underlying strategies. While some follow the EC framework (Fig. 9), similar to existing approaches, others adopt classical optimization paradigms, such as Ant Colony Optimization (ACO) in Fig. 10, Particle Swarm Optimization (PSO) in Fig. 11, Simulated Annealing in Fig. 12, Tabu Search in Fig. 13, and hybrid strategies in Fig. 15, which achieved the best performance in our evaluations. By leveraging diverse optimization principles and generating tailored prompts, MoH facilitates broader exploration of the extensive search space, enabling the discovery of more effective heuristics.

**Complexity Analysis.** Although MoH introduces an additional layer of complexity compared with previous baselines, empirical results show that it does not incur significant computational overhead (see Table 11 and 12 in Appendix). In terms of the efficiency of the generated heuristics, we also conduct additional comparisons with classical solvers (i.e., Concorde (app, 2003) and OR-Tools (Furnon & Perron, 2023)) as well as lightweight learning-based solvers (i.e., LEHD (Luo et al., 2023), SIL (Luo et al., 2024), and NeuOpt (Ma et al., 2023)) under (approximately) the same computational budget in Table 13 in Appendix. In general, our approach offers a more favorable trade-off between computational cost and performance across nearly all problem sizes.

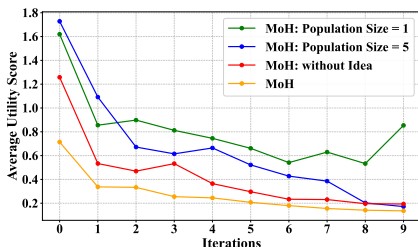

Figure 4: Training convergence curves under different settings.

## 5 CONCLUSION

We propose a novel MoH framework, which leverages LLMs to generate effective meta-optimizers for improving COP heuristics. MoH extends the heuristic design paradigm by incorporating an outer loop for heuristic-optimizer design and employs a multi-task scheme to improve generalization and enable broader heuristic exploration. Experimental results demonstrate that heuristics discovered by MoH outperform both classical heuristics and existing LLM-based approaches. We believe MoH offers a new perspective on generating promising heuristics, with the potential to surpass human-designed ones in solving NP-hard COPs. We acknowledge certain limitations of MoH, such as the search efficiency. The outer-loop and multi-task optimization may introduce additional computational overhead, highlighting the need for more efficient search strategies. Additionally, while our current scope focuses on classical COPs, MoH has the potential to address a broader range of COPs and even other classes of optimization problems, which we leave for future work.

ACKNOWLEDGMENTS

This research is supported by the National Research Foundation, Singapore under its AI Singapore AI Research Fundamental Research Collaborative (US-NSF Researcher Call) (AISG Award No: AISG3-RP-2025-036-USNSF) and AI Singapore Programme (AISG Award No: AISG3-RP-2022-031). Any opinions, findings and conclusions or recommendations expressed in this material are those of the author(s) and do not reflect the views of National Research Foundation, Singapore. This research is also supported by National Natural Science Foundation of China No. 62473233. We would like to thank the anonymous reviewers and (S)ACs of ICLR 2026 for their constructive comments and service to the community.

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

APPENDIX

## A    FREQUENTLY ASKED QUESTIONS

**Comparison of MoH with other Optimization Frameworks.**  The meta-optimization of our framework also resembles other optimization methods such as Multi-Task Optimization (Gupta et al., 2015; Osaba et al., 2022), Multi-Objective Optimization (Gunantara, 2018) and Bi-level Optimization (Xu et al., 2024). However, we emphasize our method's distinctions below:

- Traditional multi-task algorithms are explicitly designed to train a unified model capable of solving multiple tasks simultaneously, typically via parameter sharing, knowledge transfer and genetic operators (Gupta et al., 2015). In such methods, cross-task generalization is learned during training. In contrast, we build MoH on a pre-trained LLM that already exhibits broad task-solving capability, requiring no additional training of the model itself. Moreover, MoH focuses on generating improved heuristics by employing diverse optimization strategies to explore the LLM search space. In LLM-based heuristic generation, integrating evolutionary multitasking (Osaba et al., 2022) (EMT) is constrained by the semantic variability of LLM-generated code and prompts: in practice, EMT can operate only at the heuristic layer—i.e., searching over task-specific heuristics across downstream tasks—thereby reverting to a fixed LLM–EC pipeline. In contrast, our approach treats each task individually and does not assume EC as the underlying heuristic-optimizer. The meta-optimizer adaptively discovers, modifies, and even composes optimization strategies (EC or otherwise) during training, yielding a more flexible alternative that steps beyond previous LLM-EC design.

- As for multi-objective optimization (Gunantara, 2018), it is typically employed to address conflicting objectives by identifying Pareto-optimal solutions that reflect trade-offs among them. However, this paradigm is also not well-suited to our motivation, which is to evolve optimizers rather than heuristics, using LLMs for CO.

- For multi-level frameworks, our problem does indeed align with bi-level optimization structures (Xu et al., 2024), as both involve a two-stage optimization process. However, what distinguishes our work is the application of a bi-level optimization perspective to the domain of LLM-based code generation for heuristic design. This is not a straightforward extension, as it introduces several unique challenges: First, unlike typical bi-level problems where objectives and variables are clearly defined, both of our optimizer and heuristic are LLM-generated programs, whose implicit, program-level behavior (e.g., heuristic logic as code) must be evaluated by execution rather than closed-form analysis, thereby introducing additional abstraction and complexity to the problem. Second, bi-level optimization typically focuses on the co-optimization of both the upper and lower levels, where the two problems are often tightly coupled. In contrast, our approach primarily aims to improve the heuristics generated in the inner loop by exploring diverse optimization strategies in the outer loop. Moreover, the downstream tasks in our framework are flexible and can be adapted based on specific needs, making the approach more versatile across different types of problems.

In summary, while some of the mentioned optimization frameworks may hold potential, adapting them to the domain of LLM-based code generation for CO is non-trivial, particularly in a way that aligns with our goal of evolving optimizers without introducing significant complexities or computational overhead.

**Relationship of Meta-Optimizer and Heuristic-Optimizer.** In our framework, meta-optimizer and heuristic-optimizer share the same functional structure (see Fig. 3) and meta-optimizer is derived from heuristic-optimizers, their fundamental difference lies in their hierarchical roles and optimization targets within our bilevel optimization framework: 1) **Meta-optimizer (Outer Loop)**: Operates at a higher abstraction level. Its goal is to design heuristic-optimizers. It takes a population of heuristic-optimizers as input and evolves them to find better optimization strategies. 2) **Heuristic-optimizer (Inner Loop)**: Operates at the task level. Its goal is to solve specific downstream CO tasks. It takes a population of heuristics as input and evolves them to minimize the optimality gap for downstream problems (e.g., TSP, BPP). Despite these distinct roles, we implement both using a unified functional template (see 3) to enable recursive self-improvement. This allows the same LLM-generated function structure to adapt its behavior based on the specific inputs it receives. To be specific, 1) when acting as a Heuristic-Optimizer: population: Stores candidate heuristics for each task. utility: Evaluates task performance (e.g., optimality gap, cost). subtask: Specifies the task type and size (e.g., "TSP20");

2) when acting as a Meta-Optimizer: population: Stores candidate heuristic-optimizers. utility: Aggregated performance across all downstream tasks. subtask: Is set to "Optimizer", directing the function to perform meta-level optimization (as shown in Algorithm 1).

**Comparison with Meta-Prompt Optimization.** We would like to clarify that characterizing MoH as merely a "sophisticated meta-prompt optimization" may not fully capture the fundamental algorithmic shift introduced by our framework. The key distinction lies in moving beyond optimizing with a static strategy toward designing dynamic, executable search strategies. Our focus is on generating algorithmic code via LLM, not on "prompting prompts". We first clarify that our meta-optimizer is more than a meta-prompt. Concretely, meta-prompt optimization focuses on tuning textual inputs to elicit a better response. In contrast, the MoH meta-optimizer functions as an active algorithmic controller. It does not simply "ask" the LLM; it maintains a stateful process that includes population management, iterative code refinement, and feedback loops. These are structured algorithmic components that exist outside the prompt itself. This unique active algorithmic controller is formalized in our bilevel framework. Unlike prior works that rely on a single-level prompt template, MoH introduces an outer loop that dynamically searches for the optimization strategy itself. This allows MoH to explore the space of algorithms rather than just the space of prompts, a direction largely underexplored in LLM-based combinatorial optimization.

**Generalizability Claim.** The main focus of this paper is cross-size generalization, as highlighted in the abstract, introduction, and experimental sections. While our framework can, in principle, be extended to support other forms of generalization, doing so would require addressing additional challenges, which we leave for future work.

Table 6: Results for Cross-Problem Training

| Training Task | Gap |
|---|---|
| TSP-GLS-Size200 | 0.384%±0.038% |
| TSP-KGLS-Size200 | 0.219%±0.023% |
| BPP-online-5k item, cap. 100 | 0.978%±0.218% |

We will make this scope and positioning clearer in the revised version. 1) **Cross-problem generalization.** We agree that MoH does not directly generalize a heuristic learned for one CO problem to a completely different CO problem. This limitation is also shared by all baselines we compare against. Although our framework introduces a multi-task training mechanism that can, in principle, incorporate multiple problem settings, using a single trained meta-optimizer across distinct CO problems may not yield strong performance, particularly due to the substantial structural differences (e.g., objective, constraint) among CO problems. Nevertheless, we conducted a preliminary experiment in which MoH was jointly trained on TSP-GLS, TSP-KGLS, and BPP-online tasks. The results (mean±standard error across 5 runs) are shown in Table 6. We leave further improvements to future work. 2) **Cross-distribution generalization.** In fact, switching to a cross-distribution setting does not modify the heuristic's internal structure, it merely changes the distribution of the instances used for evaluation. Because the heuristic design remains intact, our method can readily generalize across distributions, and it consistently delivers strong performance under such cross-distribution settings. We evaluate cross-distribution generalization in Tables 8 and 9 using TSPLIB, whose instances do not follow the uniform distribution used during training. We also evaluate the generalization ability of the meta-optimizer trained on TSP200-Uniform by leveraging it to optimize heuristics for the TSP200-Cluster task. Table 7 shows the in-distribution (TSP200-Uniform) and cross-distribution (TSP200-Cluster) performance, respectively.

**Utility Function.** Our rationale for using size-weighted utility is that CO heuristics typically degrade more significantly as problem size increases. Thus, we place greater emphasis on larger instances to encourage the meta-optimizer to discover heuristics that remain effective at scale, an effect that is empirically supported by the table below. This weighting scheme highlights performance on more challenging problems.

Table 7: Generalization results from Uniform to Cluster TSP

| TSP-GLS | TSP200-Uniform | TSP200-Cluster |
|---|---|---|
| ReEvo | 0.331% | 0.290% |
| HSEvo | 0.328% | 0.315% |
| MoH | **0.291%** | **0.268%** |

**Novelty of Optimizer.** We do not emphasize the novelty of the meta-optimizer itself, since some of the generated meta-optimizers indeed resemble classical optimization strategies. In contrast to

approaches that typically rely on a fixed evolutionary computation (EC) heuristic-optimizer while leveraging LLMs to design heuristics for combinatorial optimization problems (COPs), our approach enables the automatic design of heuristic-optimizers that evolve CO heuristics from scratch, offering fresh insights into this field. By applying different optimization strategies within those generated heuristic-optimizers, we can explore a more diverse heuristic search space, thereby improving the performance of the discovered heuristics.

## B    RELATED WORK

Traditional heuristic design for NP-hard COPs relies heavily on expert knowledge and is time-consuming to develop (Dréo, 2006). This has motivated the emergence of automatic heuristic design (Burke et al., 2013) as a more efficient alternative (Pillay & Qu, 2021), leveraging metaheuristic or ML techniques to automate heuristic generation and optimization (Burke et al., 2007; Hutter et al., 2009; Blot et al., 2016; Mirshekarian & Sormaz, 2018). However, these approaches are often constrained by inflexible search and strong domain-specific dependencies (Ochoa et al., 2012; Branke et al., 2015). Recently, neural solvers have gained attention as a promising alternative (Vinyals et al., 2015; Kool et al., 2018), employing deep learning to learn heuristics in a data-driven manner. Despite showing promise, they still face several challenges, including limited scalability and generalization, as well as high training overhead. More recently, the advent of LLMs has transformed the landscape of heuristic design. Their advanced language understanding and reasoning capabilities (Brown et al., 2020; Wei et al., 2022; 2021) have been increasingly exploited to enhance heuristic generation for solving COPs (Liu et al., 2024b; Romera-Paredes et al., 2024; Sun et al., 2024). Among recent efforts, most approaches combine the efficiency of evolutionary search with the adaptability of LLM reasoning via few-shot prompting, leading to a surge of interest in LLM-EC frameworks for heuristic design in COPs (Liu et al., 2024a; Ye et al., 2024; Yao et al., 2024). However, the use of a fixed optimization strategy (e.g., EC) in these frameworks often restricts exploration of the broader search space. In addition to serving as heuristic generators, LLMs have also been employed to directly generate solutions or formulate mathematical models for solving COPs. In the following, we provide a comprehensive review of neural and LLM-based approaches.

### B.1    NEURAL HEURISTICS FOR COPS

Unlike traditional hand-crafted heuristics, neural heuristics for solving COPs have rapidly advanced in recent years (Bengio et al., 2021; Berto et al., 2025a). These methods generally fall into two paradigms. 1) For constructive heuristics, Pointer Network (Ptr-Net) (Vinyals et al., 2015), a sequence-to-sequence model with differentiable attention mechanisms, was first introduced to directly learn permutation-invariant solutions for TSP through supervised learning. This was extended by using reinforcement learning to improve performance (Bello et al., 2016), and further applied to CVRP (Nazari et al., 2018). With the rise of Transformer architectures (Vaswani, 2017), the attention-based model (Kool et al., 2018) was proposed to solve various COPs, inspiring a series of subsequent works (Kim et al., 2021; Kwon et al., 2020; Drakulic et al., 2023; Luo et al., 2023; Zhou et al., 2023; Bi et al., 2024). More recently, there has been a surge of interest in foundation models that aim to solve multiple COPs using a single, general-purpose model (Zhou et al., 2024; Berto et al., 2025b; Drakulic et al., 2025). 2) Improvement heuristics (Hottung & Tierney, 2020; Wu et al., 2021; Li et al., 2023; Ma et al., 2023; Sun & Yang, 2023) leverage neural networks to guide local search for solution refinement (Hudson et al., 2021; Sui et al., 2024). While these approaches can often produce (near-)optimal solutions with extended inference times, they typically face challenges in scaling to large problem instances and generalizing across diverse problem settings.

### B.2    LLMS FOR COPS

LLMs have recently gained widespread recognition and found broad applications across various domains (Ji et al., 2023; Kaddour et al., 2023), significantly influencing research directions in combinatorial optimization. In particular, recent studies have explored the application of LLMs in multiple facets of CO, including enhancing algorithm design (Liu et al., 2024a; Romera-Paredes et al., 2024; Ye et al., 2024), automating the formulation of CO problems (AhmadiTeshnizi et al., 2024; Jiang et al., 2025; Li et al., 2024; Xiao et al., 2023), developing CO-specific benchmark datasets (Fan et al., 2023; Sun et al., 2025), directly solving COPs (Abgaryan et al., 2024; Iklassov et al., 2024), and

integrating LLMs into domain-specific foundation models to construct unified frameworks capable of addressing a wide spectrum of CO tasks (Andreychuk et al., 2025; Jiang et al., 2024b). As LLMs continue to evolve rapidly, they exhibit great potential to support the development of more automated, generalizable, and efficient problem-solving frameworks in the field of CO.

# C    ADDITIONAL RESULTS

## C.1    RESULTS ON TSPLIB

We further evaluate our method on the widely used TSPLib dataset across various instance sizes. As shown in Table 8 and 9, we compare the TSP-GLS and TSP-KGLS settings against EoH (Liu et al., 2024a), ReEvo (Ye et al., 2023), HSEvo (Dat et al., 2024), MCTS-AHD (Zheng et al., 2025), Neural Combinatorial Solvers (Kool et al., 2018; Kwon et al., 2020; Luo et al., 2023; 2024) and GLS algorithms (Hudson et al., 2021; Sui et al., 2024; Voudouris et al., 2010; Shi et al., 2018; Arnold & Sörensen, 2019) across instances of different scales. The results are split into two tables: one for instances smaller than 200, and another for sizes ranging from 200 to 1000. To handle the distributional diversity in TSPLIB, we adopt an instance-level heuristic selection strategy. Instead of using a single "best" heuristic from Table 1, MoH evaluates all size-specific best heuristics trained on the uniform distribution and selects the best-performing one for each TSPLIB instance. This allows MoH to adapt to the heterogeneous characteristics of TSPLIB instances. For fairness, all baselines follow the same instance-wise selection procedure, where each method chooses its best-performing heuristic among its size-specific candidates. Consistent with the results in Table 1, our generated heuristics outperform both GLS and KGLS baselines on most TSPLib instances. Heuristic 2 and 3 show examples of the best heuristics generated for the TSP-GLS and TSP-KGLS settings of size 1000, respectively.

Table 8: Results on TSPLib instances with sizes smaller than 200.

| Instance | Neural Solver | | | | GLS Algorithms | | | | | TSP-GLS | | | TSP-KGLS | | | |
|---|---|---|---|---|---|---|---|---|---|---|---|---|---|---|---|---|
| | AM | POMO | LEHD | SIL | GNNGLS | NeuralGLS | GLS | EBGLS | KGLS | EoH | HSEvo | MoH | ReEvo | HSEvo | MCTS-AHD | MoH |
| eil51 | 1.63 | 0.83 | 1.64 | 0.67 | 0.00 | 0.00 | 0.67 | 0.67 | 0.67 | 0.67 | 0.67 | 0.67 | 0.67 | 0.67 | 0.67 | 0.67 |
| berlin52 | 4.17 | 0.04 | 0.03 | 0.03 | 0.14 | 0.00 | 0.03 | 0.03 | 0.03 | 0.03 | 0.03 | 0.03 | 0.03 | 0.03 | 0.03 | 0.03 |
| st70 | 1.74 | 0.31 | 0.33 | 0.31 | 0.76 | 0.00 | 0.31 | 0.31 | 0.31 | 0.31 | 0.31 | 0.31 | 0.31 | 0.31 | 0.31 | 0.31 |
| eil76 | 1.99 | 1.18 | 2.54 | 1.18 | 0.16 | 0.00 | 1.37 | 1.18 | 1.18 | 1.18 | 1.18 | 1.18 | 1.18 | 1.18 | 1.18 | 1.18 |
| pr76 | 0.82 | 0.00 | 0.22 | 0.00 | 0.04 | 0.82 | 0.00 | 0.00 | 0.00 | 0.00 | 0.00 | 0.00 | 0.00 | 0.00 | 0.00 | 0.00 |
| rat99 | 2.65 | 2.39 | 1.10 | 0.73 | 0.55 | 0.72 | 1.55 | 0.74 | 0.68 | 0.68 | 0.68 | 0.68 | 0.68 | 0.68 | 0.68 | 0.68 |
| kroA100 | 4.02 | 0.41 | 0.12 | 0.02 | 0.73 | 0.03 | 0.02 | 0.02 | 0.06 | 0.02 | 0.02 | 0.02 | 0.02 | 0.02 | 0.02 | 0.02 |
| kroB100 | 5.14 | 0.32 | 0.26 | 0.00 | 0.15 | 0.88 | 0.23 | 0.00 | 0.25 | 0.00 | 0.00 | 0.00 | 0.00 | 0.00 | 0.00 | 0.00 |
| kroC100 | 0.97 | 0.18 | 0.32 | 0.01 | 1.57 | 1.77 | 0.50 | 0.01 | 0.01 | 0.01 | 0.01 | 0.01 | 0.01 | 0.01 | 0.01 | 0.01 |
| kroD100 | 2.72 | 0.84 | 0.38 | 0.00 | 0.57 | 0.00 | 0.00 | 0.20 | 0.00 | 0.00 | 0.00 | 0.00 | 0.00 | 0.00 | 0.00 | 0.00 |
| kroE100 | 1.47 | 0.45 | 0.43 | 0.17 | 1.22 | 1.05 | 0.49 | 0.00 | 0.07 | 0.00 | 0.00 | 0.00 | 0.00 | 0.00 | 0.00 | 0.00 |
| rd100 | 3.41 | 0.01 | 0.01 | 0.01 | 0.46 | 0.00 | 0.01 | 0.01 | 0.02 | 0.01 | 0.01 | 0.01 | 0.01 | 0.01 | 0.01 | 0.01 |
| eil101 | 2.99 | 1.84 | 2.31 | 2.07 | 0.20 | 0.36 | 3.28 | 1.91 | 2.07 | 1.78 | 1.82 | 1.78 | 1.78 | 1.78 | 1.78 | 1.78 |
| lin105 | 1.74 | 0.52 | 0.34 | 0.03 | 0.61 | 0.65 | 0.03 | 0.03 | 0.03 | 0.03 | 0.03 | 0.03 | 0.03 | 0.03 | 0.03 | 0.03 |
| pr107 | 3.93 | 0.52 | 11.24 | 0.00 | 0.44 | 0.81 | 0.40 | 0.00 | 0.00 | 0.00 | 0.00 | 0.00 | 0.00 | 0.00 | 0.00 | 0.00 |
| pr124 | 3.68 | 0.60 | 1.11 | 0.00 | 0.76 | 0.08 | 0.60 | 0.60 | 0.08 | 0.00 | 0.00 | 0.00 | 0.00 | 0.00 | 0.00 | 0.00 |
| bier127 | 5.91 | 13.72 | 4.76 | 0.01 | 1.95 | 2.73 | 0.59 | 0.29 | 0.42 | 0.01 | 0.01 | 0.01 | 0.01 | 0.04 | 0.10 | 0.01 |
| ch130 | 3.18 | 0.16 | 0.55 | 0.25 | 3.52 | 1.19 | 1.09 | 0.46 | 0.01 | 0.01 | 0.01 | 0.01 | 0.01 | 0.01 | 0.01 | 0.01 |
| pr136 | 5.06 | 0.93 | 0.45 | 0.02 | 3.39 | 2.32 | 2.01 | 0.28 | 0.24 | 0.00 | 0.00 | 0.00 | 0.00 | 0.01 | 0.00 | 0.00 |
| pr144 | 7.64 | 0.53 | 0.19 | 0.09 | 3.58 | 0.74 | 0.09 | 0.00 | 0.00 | 0.00 | 0.00 | 0.00 | 0.00 | 0.00 | 0.00 | 0.00 |
| ch150 | 4.58 | 0.53 | 0.52 | 0.04 | 2.11 | 2.49 | 0.68 | 0.37 | 0.04 | 0.04 | 0.33 | 0.04 | 0.04 | 0.04 | 0.04 | 0.04 |
| kroA150 | 3.78 | 0.70 | 1.40 | 0.00 | 2.98 | 0.77 | 1.75 | 0.26 | 0.17 | 0.00 | 0.00 | 0.00 | 0.00 | 0.00 | 0.00 | 0.00 |
| kroB150 | 2.44 | 1.17 | 0.76 | 0.00 | 3.26 | 3.11 | 1.01 | 0.00 | 0.08 | 0.00 | 0.00 | 0.00 | 0.00 | 0.00 | 0.00 | 0.00 |
| pr152 | 7.49 | 1.05 | 12.14 | 0.19 | 3.12 | 0.00 | 0.19 | 0.19 | 0.19 | 0.00 | 0.00 | 0.00 | 0.19 | 0.19 | 0.19 | 0.00 |
| u159 | 7.55 | 0.95 | 1.13 | 0.00 | 1.02 | 0.90 | 0.74 | 0.78 | 0.96 | 0.00 | 0.00 | 0.00 | 0.00 | 0.00 | 0.00 | 0.00 |
| rat195 | 6.89 | 8.15 | 1.42 | 0.47 | 1.67 | 0.48 | 0.61 | 0.61 | 0.97 | 0.90 | 0.80 | 0.56 | 0.65 | 0.47 | 0.61 | 0.47 |
| d198 | 373.02 | 17.29 | 9.23 | 0.46 | 4.77 | 1.28 | 2.08 | 1.87 | 0.31 | 0.32 | 0.26 | 0.21 | 0.20 | 0.28 | 0.43 | 0.20 |
| kroA200 | 7.11 | 1.58 | 0.64 | 0.00 | 2.03 | 0.86 | 0.75 | 0.18 | 0.71 | 0.13 | 0.04 | 0.00 | 0.23 | 0.09 | 0.04 | 0.00 |
| kroB200 | 8.54 | 1.44 | 0.16 | 0.01 | 2.59 | 3.74 | 1.43 | 1.27 | 0.89 | 0.08 | 0.05 | 0.04 | 0.01 | 0.01 | 0.01 | 0.01 |
| Average | 16.77 | 2.02 | 1.92 | 0.23 | 1.53 | 0.96 | 0.78 | 0.42 | 0.36 | 0.21 | 0.22 | **0.19** | 0.21 | 0.20 | 0.21 | **0.19** |

## C.2    COST AND EVALUATION COMPARISON

We present a comprehensive cost comparison across methods by reporting their average computational metrics, i.e., the LLM request counts, token usage, evaluation numbers, and performance, for the two studied problem settings: TSP-GLS (Table 10 and 11) and CVRP (Table 12). For TSP-GLS, we use instances of size 100 and 200 for both training and inference to ensure fair comparison between MoH and baseline methods. For CVRP, training involves instance sizes of 20 and 50. All results are averaged over three runs, with token usage evaluated using the GPT-4o-mini API.

From the comparison between Table 10 and 11, we want to highlight that for the same problem type, the time cost is determined solely by the problem size. This is because the overall runtime is dominated by the evaluation phase—i.e., executing the generated heuristics to compute their scores.

Table 9: Results on TSPLib instances with sizes ranging from 200 to 1000.

| Instance | Neural Solver | | | TSP-GLS | | | TSP-KGLS | | | |
| --- | --- | --- | --- | --- | --- | --- | --- | --- | --- | --- |
| | POMO | LEHD | SIL | EoH | HSEvo | MoH | ReEvo | HSEvo | MCTS-AHD | MoH |
| ts225 | 3.60 | 0.28 | 0.00 | 0.00 | 0.00 | 0.00 | 0.00 | 0.00 | 0.00 | 0.00 |
| tsp225 | 3.17 | 0.00 | 0.00 | 0.00 | 0.00 | 0.00 | 0.00 | 0.00 | 0.00 | 0.00 |
| pr226 | 1.42 | 1.11 | 0.04 | 0.00 | 0.04 | 0.00 | 0.00 | 0.00 | 0.00 | 0.00 |
| pr264 | 2.80 | 5.48 | 0.00 | 0.00 | 0.00 | 0.00 | 0.00 | 0.00 | 0.00 | 0.00 |
| a280 | 5.23 | 3.02 | 0.30 | 0.55 | 0.80 | 0.30 | 0.30 | 0.30 | 0.30 | 0.30 |
| pr299 | 4.94 | 2.81 | 0.01 | 0.07 | 0.53 | 0.11 | 0.13 | 0.07 | 0.08 | 0.01 |
| lin318 | 4.72 | 1.41 | 0.69 | 0.55 | 1.13 | 0.38 | 0.32 | 0.27 | 0.43 | 0.29 |
| rd400 | 6.37 | 1.00 | 0.00 | 0.78 | 0.82 | 0.29 | 0.17 | 0.38 | 0.45 | 0.20 |
| fl417 | 8.51 | 7.76 | 2.42 | 0.64 | 0.68 | 0.62 | 0.60 | 0.49 | 0.70 | 0.49 |
| pr439 | 7.87 | 3.37 | 0.01 | 1.09 | 2.46 | 0.28 | 1.01 | 1.21 | 1.23 | 0.56 |
| pcb442 | 5.36 | 3.11 | 0.04 | 0.59 | 1.12 | 0.79 | 0.28 | 0.35 | 0.15 | 0.08 |
| d493 | 9.67 | 9.49 | 0.24 | 1.12 | 1.11 | 0.51 | 0.61 | 0.67 | 1.42 | 0.45 |
| u574 | 11.86 | 2.73 | 0.28 | 1.12 | 0.87 | 0.80 | 0.86 | 1.49 | 1.39 | 0.83 |
| rat575 | 12.46 | 3.02 | 0.85 | 2.55 | 1.25 | 1.35 | 1.67 | 1.60 | 1.47 | 1.04 |
| p654 | 11.30 | 3.30 | 2.77 | 0.21 | 0.31 | 0.12 | 0.15 | 1.57 | 0.11 | 0.10 |
| d657 | 12.72 | 8.05 | 1.00 | 1.42 | 1.26 | 0.87 | 1.02 | 1.08 | 0.96 | 0.54 |
| u724 | 16.57 | 3.27 | 0.19 | 1.06 | 2.07 | 0.99 | 0.91 | 0.92 | 0.88 | 0.72 |
| rat783 | 18.11 | 3.91 | 0.65 | 2.46 | 1.98 | 2.18 | 1.68 | 1.65 | 1.86 | 1.11 |
| pr1002 | 20.00 | 4.44 | 0.51 | 1.83 | 1.11 | 1.14 | 1.21 | 1.27 | 1.17 | 0.90 |
| Average | 8.77 | 3.56 | 0.53 | 0.84 | 0.92 | 0.57 | 0.57 | 0.70 | 0.66 | **0.40** |

Running a heuristic on a larger instance (e.g., 200 nodes) naturally requires more CPU time than on a smaller instance (e.g., 100 nodes), affecting all methods equally. In contrast, token consumption and LLM request counts are primarily dictated by the algorithmic design and prompting strategy of different problem settings, rather than by the problem size. The LLM's role is to generate heuristic code (i.e., algorithmic logic), which is inherently agnostic to the scale of the specific problem instance. This claim is also empirically supported by the comparison between Table 11 and Table 12.

Table 10: Cost of different methods on TSP-GLS setting with size 100.

| Methods | Time/mins | Input Tokens | Output Tokens | Total Tokens | LLM requests | Evaluation count | Results-Gap |
| --- | --- | --- | --- | --- | --- | --- | --- |
| MoH Train | 196.9 | 927537.0 | 589250.3 | 1516787.3 | 1347.7 | 989.3 | 0.034% |
| MoH Inference | 46.7 | 526404.3 | 169028.3 | 695432.6 | 311.3 | 282.7 | 0.036% |
| EoH | 223.2 | 1282802.0 | 390085.5 | 1672887.5 | 1119.0 | 1001.7 | 0.055% |
| HSEvo | 214.7 | 1291765.3 | 297487.0 | 1589252.3 | 691.3 | 1015.0 | 0.071% |

Table 11: Cost of different methods on TSP-GLS setting with size 200.

| Methods | Time/mins | Input Tokens | Output Tokens | Total Tokens | LLM requests | Evaluation budget | Results-Gap |
| --- | --- | --- | --- | --- | --- | --- | --- |
| MoH-Train | 238.4 | 743837.0 | 458358.7 | 1202195.7 | 1248.3 | 971.7 | 0.373% |
| MoH-Inference | 62.7 | 398064.3 | 138350.0 | 536414.3 | 276.7 | 240.3 | 0.398% |
| EoH | 326.4 | 1437973.0 | 446583.7 | 1884556.7 | 1256.0 | 992.0 | 0.535% |
| HSEvo | 291.7 | 1188412.3 | 390665.0 | 1579077.3 | 679.7 | 1005.0 | 0.448% |

Table 12: Cost of different methods on CVRP+ACO setting.

| Methods | Time/mins | Input Tokens | Output Tokens | Total Tokens | LLM requests | Evaluation budget | Results-Obj. 20 | Results-Obj. 50 |
| --- | --- | --- | --- | --- | --- | --- | --- | --- |
| MoH-train | 321.1 | 986216.0 | 740292.3 | 1726508.3 | 1456.7 | 938.3 | 4.831 | 9.262 |
| MoH-inference | 107.5 | 479618.0 | 153284.3 | 632902.3 | 330.3 | 307.3 | 4.837 | 9.254 |
| ReEvo | 231.2 | 2122802.0 | 677652.0 | 2800454.0 | 1431.0 | 1000.0 | 4.877 | 9.521 |
| HSEvo | 350.7 | 2140030.7 | 670590.0 | 2810620.7 | 1076.7 | 1001.7 | 4.977 | 9.536 |
| MCTS | 1330.3 | 2601902.0 | 825037.3 | 3426939.3 | 1490.0 | 1002.7 | 4.881 | 9.233 |

## C.3 COMPARISON WITH CLASSICAL AND LEARNING-BASED SOLVERS

We also make additional comparison with classical solvers (i.e., Concorde and OR-Tools) as well as lightweight learning-based solvers (i.e., LEHD (Luo et al., 2023), SIL (Luo et al., 2024), and NeuOpt (Ma et al., 2023)) under (approximately) the same computational budget in Table 13. Specifically, we control the solving time across all methods to be comparable to the inference time of MoH, except for Concorde, whose runtime cannot be constrained. Classical solvers often suffer from scalability issues due to the NP-hard nature of CO problems, while learning-based solvers face generalization challenges (e.g., NeuOpt struggles to generalize to larger instances and SIL struggles in generalization to smaller sizes). In contrast, our approach offers a more favorable trade-off between computational cost and performance across nearly all problem sizes. This highlights the practical value of MoH in real-world scenarios where routine, varied-scale CO solving is required, as the task-level deployment setting more accurately reflects practical applications of heuristic design and justifies the initial computational investment.

Table 13: Optimality gap(%) and averaging solving time(s) of instances of different problem sizes across different baselines.

| Problem Size | 100 | | 200 | | 500 | | 1000 | |
|---|---|---|---|---|---|---|---|---|
| Methods | Gap | Average Time | Gap | Average Time | Gap | Average Time | Gap | Average Time |
| Concorde | 0.000% | 0.260s | 0.000% | 1.094s | 0.000% | 11.878s | 0.000% | 229.528s |
| OR-Tools | 2.529% | 2.024s | 3.843% | 3.001s | 4.751% | 10.007s | 5.001% | 45.099s |
| LEHD | 0.375% | 1.086s | 0.446% | 2.709s | 0.792% | 9.066s | 1.680% | 44.469s |
| SIL | 4.073% | 1.122s | 2.545% | 2.159s | 1.459% | 8.789s | 1.051% | 43.025s |
| NeuOpt | 0.471% | 1.432s | 0.414% | 4.411s | 125.864% | 45.314s | - | - |
| MoH-GLS | 0.012% | 1.075s | 0.291% | 2.335s | 0.936% | 7.818s | 1.476% | 21.681s |
| MoH-KGLS | 0.002% | 0.394s | 0.177% | 1.497s | 0.805% | 10.866s | 1.365% | 45.518s |

## C.4 COMPREHENSIVE COMPARISON OF BASELINES ON DIFFERENT LLMS

To provide a broader evaluation, we compare multiple LLMs across different baselines and problem sizes. This allows us to verify the consistency of performance trends beyond a single setting. Table 14 reports results for GPT 4o-mini, o1-mini, DeepSeek-v3, and Qwen-plus combined with EoH, HSEvo, and MoH over problem sizes 100–1000. Each entry shows mean error with standard deviation. Overall, MoH demonstrates clear superiority across settings. Compared to EoH and HSEvo, MoH achieves consistently lower errors, particularly on larger problem sizes (500 and 1000).

Table 14: Optimality gaps (mean% ± std%) of baselines on different LLMs across 5 runs.

| Baselines | Problem Size | GPT 4o-mini | o1-mini | DeepSeek-v3 | Qwen-plus |
|---|---|---|---|---|---|
| EoH | 100 | 0.051%±0.013% | 0.052%±0.010% | 0.072%±0.030% | 0.195%±0.142% |
| | 200 | 0.426%±0.043% | 0.439%±0.032% | 0.471%±0.087% | 0.780%±0.289% |
| | 500 | 1.927%±0.491% | 1.486%±0.145% | 1.692%±0.482% | 2.007%±0.535% |
| | 1000 | 3.323%±0.516% | 2.857%±0.271% | 3.362%±0.355% | 3.485%±0.295% |
| HSEvo | 100 | 0.076%±0.066% | 0.042%±0.011 | 0.042%±0.026% | 0.044%±0.022% |
| | 200 | 0.729%±0.310% | 0.441%±0.020 | 0.590%±0.236% | 0.497%±0.168% |
| | 500 | 1.726%±1.206% | 1.811%±0.091 | 1.842%±0.605% | 1.755%±0.537% |
| | 1000 | 3.261%±0.775% | 3.493%±0.222 | 3.006%±0.525% | 2.797%±0.430% |
| MoH | 100 | 0.031%±0.020% | 0.024±0.006% | 0.080±0.065% | 0.040±0.011% |
| | 200 | 0.345%±0.039% | 0.364±0.011% | 0.392%±0.027% | 0.387%±0.018% |
| | 500 | 1.187%±0.280% | 1.353%±0.244% | 1.478%±0.340% | 1.625%±0.354% |
| | 1000 | 1.889%±0.308% | 2.332%±0.298% | 2.669%±0.310% | 3.283%±0.218% |

## C.5 STATISTICAL SIGNIFICANCE ANALYSIS

To further validate the robustness of our results, we conduct statistical significance tests comparing MoH against a wide range of baselines under both TSP constructive and TSP-KGLS settings. As

shown in Table 15, we conduct one-sided Wilcoxon signed-rank tests on the objective values of the heuristics generated by MoH and the corresponding baselines. The results are averaged from three best runs of each method. The results demonstrate that our method consistently achieves statistically significant improvements over all baselines. In particular, all p-values are far below the 0.05 threshold, providing strong evidence that the observed gains are not due to random chance. This indicates that the observed improvements of heuristic performance by MoH are not due to random chance but are statistically significant.

Table 15: One-sided Wilcoxon signed-rank test p-values comparing MoH with baselines.

| Problem | TSP constructive | | | | | TSP-KGLS | | |
|---------|-----------|--------|--------|--------|----------|--------|--------|----------|
| Baselines | FunSearch | EoH | ReEvo | HsEvo | MCTS-AHD | ReEvo | HsEvo | MCTS-AHD |
| MoH-200 | 2.73E-55 | 1.48E-15 | 1.61E-03 | 3.55E-22 | 8.45E-12 | 5.30E-03 | 1.14E-02 | 1.59E-04 |
| MoH-500 | 2.12E-60 | 1.49E-19 | 4.86E-09 | 8.75E-46 | 3.56E-27 | 3.90E-11 | 7.12E-15 | 4.05E-07 |
| MoH-1000 | 2.40E-64 | 1.61E-34 | 2.25E-31 | 1.70E-64 | 5.26E-34 | 1.01E-46 | 2.78E-36 | 1.10E-08 |

## C.6 STATISTICAL PERFORMANCE

To further validate the robustness of our results, we conduct repeated evaluations under each problem size and report the mean ± standard deviation across multiple runs (see Table 16). This statistical summary demonstrates that MoH not only achieves the best average performance but also maintains low variability, confirming the reliability and stability of our proposed method.

Table 16: Statistical performance of different baselines for TSP.

| Problem Size | 20 | 50 | 100 | 200 | 500 | 1000 |
|--------------|-----|-----|-----|-----|-----|------|
| Constructive TSP | | | | | | |
| Funsearch | 12.062%±0.907% | 16.049%±1.340% | 18.000%±1.487% | 20.507%±1.757% | 21.646%±2.554% | 22.873%±2.976% |
| EoH | 10.115%±0.597% | 13.180%±0.851% | 14.485%±1.076% | 15.890%±1.000% | 16.145%±0.672% | 17.224%±0.672% |
| ReEvo | 9.519%±0.192% | 12.317%±0.269% | 13.379%±0.319% | 14.654%±0.230% | 15.412%±0.233% | 16.258%±0.194% |
| HsEvo | 10.938%±2.878% | 12.477%±1.982% | 14.105%±1.639% | 16.315%±1.423% | 18.108%±1.964% | 18.963%±2.250% |
| MCTS-AHD | 8.407%±1.140% | 12.545%±1.284% | 14.128%±1.511% | 15.727%±1.753% | 16.813%±1.454% | 17.306%±1.089% |
| MoH | **8.599%±1.032%** | **12.307%±2.019%** | **13.046%±1.540%** | **14.103%±0.476%** | **14.778%±0.469%** | **15.867%±0.511%** |
| Improvement TSP | | | | | | |
| EoH-GLS | 0.000%±0.000% | 0.000%±0.000% | 0.051%±0.013% | 0.426%±0.043% | 1.927%±0.491% | 3.323%±0.516% |
| HsEvo-GLS | 0.000%±0.000% | 0.000%±0.000% | 0.076%±0.066% | 0.729%±0.310% | 1.726%±1.206% | 3.261%±0.775% |
| ReEvo-GLS | 0.000%±0.000% | 0.000%±0.000% | 0.063%±0.027% | 0.627%±0.340% | 2.060%±0.444% | 3.491%±0.665% |
| MoH | **0.000%±0.000%** | **0.000%±0.000%** | **0.031%±0.020%** | **0.345%±0.039%** | **1.187%±0.280%** | **1.889%±0.308%** |
| ReEvo-KGLS | 0.000%±0.000% | 0.000%±0.000% | 0.005%±0.002% | 0.223%±0.002% | 0.981%±0.004% | 1.616%±0.020% |
| HsEvo-KGLS | 0.000%±0.000% | 0.000%±0.000% | 0.006%±0.002% | 0.228%±0.006% | 1.003%±0.046% | 1.667%±0.071% |
| MCTS-AHD-KGLS | 0.000%±0.000% | 0.000%±0.000% | 0.011%±0.004% | 0.239%±0.025% | 0.942%±0.062% | 1.478%±0.091% |
| MoH-KGLS | **0.000%±0.000%** | **0.000%±0.000%** | **0.004%±0.001%** | **0.200%±0.021%** | **0.891%±0.053%** | **1.474%±0.088%** |

## C.7 ADDITIONAL RESULTS ON OTHER OPTIMIZATION PROBLEMS

In principle, our method can be applied to other heuristic optimization tasks that can be represented as a function and have a corresponding evaluation metric (utility). To this end, we conducted additional experiments on the Quadratic Assignment Problem (QAP) and the Acrobot system control task (Liu et al., 2024c). For Acrobot, the reported metric is the minimum number of steps required to complete the task across 100 randomly initialized conditions. For QAP, the cost is averaged over 128 instances. The results are listed in Table 17.

Table 17: Performance comparison on Acrobot and QAP tasks.

| Task | Domain | Metric | ReEvo | MoH (Ours) |
|------|--------|--------|-------|-----------|
| **Acrobot** | Robotics Control | Min. Steps ↓ | $99.65 \pm 37.60$ | $\mathbf{88.76 \pm 14.19}$ |
| **QAP** | Facility Layout | Cost ↓ | 5,913,687.95 | **5,833,648.08** |

## D PROMPT DESIGN

### D.1 PROMPTS FOR HEURISTIC-OPTIMIZER GENERATION

We present the prompt used to format and generate the heuristic-optimizer, along with those embedded within the seed meta-optimizer to guide idea generation and code synthesis for both the heuristic-optimizer and downstream tasks, as illustrated in Fig. 5. In addition to the predefined prompt constraints, we also integrate additional judgment mechanisms into our framework to ensure the explainability and efficiency of generated hyper-heuristics, mitigating the potential impact of LLM output uncertainty on MoH performance.

---

**Prompt for Heuristic-Optimizer Generation**

**Task:** You should design an efficient metaheuristic using the following constraints. Your solution code should balance exploration and exploitation creatively.

**Firstly,** describe your meta-heuristic, including optimization strategy and main optimization steps in one sentence. The description must be inside a brace and marked as a comment.

**Next,** implement it in Python as a function named 'optimize_algorithm'. This function should accept five inputs: 'population', 'utility', 'language_model', 'subtask_prompt' and 'subtask'. The function should return three output: 'best_idea', 'best_solution', 'best_utility'. 'utility' is a function that evaluates solutions based on a score function, 'subtask_prompt' is the format for model responses, 'task' is the name of the problem to be optimized. The function returns 'best_idea','best_solution','best_utility' which are the idea behind best solution, best code together with its utility.

**Note:** 'language_model' is an instance of the language model class used for code generation, with function "def prompt_batch(expertise,message_batch,temperature), return responses_list" for multiple request to LLM and "def prompt(self, expertise, message, temperature) return result" for single request to LLM, 'population' is a dictionary of several historical best solutions of the task, you only use the following functions to operate: "def get_solution_by_index(self, task_name, index): return item", to get a solution by its utility rank; "def get_random_solution(self, task_name): return item", to get a random solution from the population; the item returned above is a dictionary with keys 'best_sol' and 'utility'. Other functions you can use are: "def get_subtask_size(self, task_name):" to return the size of the population.

---

**Prompts inside Seed Meta-Optimizer**

**1. Idea Generation:**
Given the following heuristic for task: ['best_sol'] with its idea: ['idea'] and utility score: ['utility'], " "Summarize the key idea from this heuristic, then provide several totally different or refined ideas from the given one to design improved algorithms with lower utility score. " "Provide a single string as the answer, less than 50 words. Your response should be formatted as a json structure.

**2. Heuristic Code Solution Generation:**
Improve the following solution: {selected_solution}. You must return an improved solution. Formatted as follows:{subtask_prompt}. To better solve the problem, you are encouraged to develop new solutions based on the direction proposed: {direction} You will be evaluated based on a score function. The lower the score, the better the solution is. Be as creative as you can under the constraints.

---

Figure 5: Prompts for generating the heuristic-optimizer and those within the seed version.

### D.2 PROMPTS FOR FORMULATING HEURISTIC GENERATION

In this section, we present a prompt example used to guide the generation of downstream COP heuristics, as shown in Fig. 6. For different tasks, only the function signature and corresponding problem size are modified accordingly.

### D.3 PROMPTS FOR HEURISTIC INITIALIZATION

To maintain population diversity, we initialize a population before training and retain elite solutions for idea modification during inference. Accordingly, we present the prompts used to generate diverse ideas that guide code generation at the start of each stage.

> You are an expert in TSP solver. Given a set of nodes with their coordinates, you need to find the shortest route that visits each node once and returns to the starting node. The task can be solved step-by-step by starting from the current node and iteratively choosing the next node. Help me design a novel algorithm that is different from the algorithms in literature to select the next node in each step.
> First, describe your new algorithm and main steps in one sentence. The description must be inside a brace and marked as a comment. Next, implement it in Python as a function named "select_next_node". This function should accept 4 input(s): "current_node","destination_node","univisited_nodes","distance_matrix". The function should return 1 output(s): "next_node". 'current_node', 'destination_node', 'next_node', and 'unvisited_nodes' are node IDs. 'distance_matrix' is the distance matrix of nodes. All are Numpy arrays. Do not give additional explanations. Your solution should be designed and fit for the task {prob} with problem size {size}.

Figure 6: Prompts for generating constructive heuristics for TSP.

> **1. Generate seed direction in training stage.**
> You are an expert in the domain of optimization heuristics and combinatorial optimization problems. Your task is to design heuristics that can effectively solve optimization problems.The problem is {problem} with corresponding size {size}. According to the task description: {task_description} Provide several high-level directions for generating the seed prompt, each aimed at minimizing the utility as a result. Format your response as a JSON codeblock below: {{ "direction": [ "content": "Your first direction suggestion here.", "content": "Your second direction suggestion here.", "content": "Your third direction suggestion here.", ... "content": "..." ] }}
>
> **2. Generate/Modify seed direction in inference stage.**
> You are an expert in optimization heuristics, tasked with summarizing key insights to design improved algorithms. Given the following heuristics for the problem: {problem}: {solution}, please summarize the key insights in the heuristics to design improved algorithms for larger sized problem problem with corresponding size {size}. Formatted as a json structure: "'json{{"insights":["content","content","content", ... ,"content"]}}'". Remember each insight inside the list should be one sentence less than 50 words.
>
> **3. Generate Code by Idea.**
> You are an expert in the domain of optimization heuristics and combinatorial optimization problems. Your task is to design heuristics that can effectively solve optimization problems. Write a function that will implement a Python algorithm to solve a problem as well as possible. The optimization problem is {problem} and the size you should focus on is {size}. The output function is formatted as follows:"'python{formula_str}'" You are encouraged to develop the algorithm that follows the direction: {direction}.

Figure 7: Prompts for code and idea generation during the initialization of training and inference.

# E    EXAMPLES OF LLM-GENERATED HEURISTICS AND META-OPTIMIZERS

In this section, we present the best-performing heuristics for the largest instance size of each problem in Heuristic 1-6, along with several examples of generated meta-optimizers shown in Fig 8-15. These examples demonstrate that MoH can produce diverse, explainable, and effective optimizers that extend beyond traditional LLM-EC, incorporating a wide range of optimization strategies and generating high-quality heuristics for downstream tasks. For clarity and space efficiency, non-essential code elements are omitted while preserving the core optimization logic.

```python
import numpy as np
def select_next_node(current_node, destination_node, unvisited_nodes, distance_matrix):
    num_unvisited = len(unvisited_nodes)
    if num_unvisited == 0:
        return None
    distances = distance_matrix[current_node, unvisited_nodes]
    avg_distance = np.mean(distances)
    threshold = 0.5 * avg_distance
    close_nodes = unvisited_nodes[distances <= threshold]
    scores = {}
    if len(close_nodes) > 0:
        for node in close_nodes:
            immediate_distance = distance_matrix[current_node, node]
            future_savings = np.sum(distance_matrix[node, close_nodes]) / (len(close_nodes) - 1) if
    len(close_nodes) > 1 else 0
            diversity_score = np.mean(distance_matrix[node, unvisited_nodes]) / (immediate_distance + 1)
            scores[node] = immediate_distance + (0.6 * (1 - future_savings)) - (0.4 * diversity_score)
    if not scores:
        far_nodes = unvisited_nodes[distances > threshold]
        for node in far_nodes:
            scores[node] = distance_matrix[current_node, node]
    next_node = min(scores, key=scores.get) if scores else None
    return next_node
```

Heuristic 1: Best constructive heuristic discovered for TSP with size 1000.

```python
import numpy as np
def update_edge_distance(edge_distance, local_opt_tour, edge_n_used):
    updated_edge_distance = np.copy(edge_distance)
    num_nodes = len(local_opt_tour)
    window_size = 5
    for i in range(num_nodes):
        current_city = local_opt_tour[i]
        for j in range(1, window_size + 1):
            next_index = (i + j) % num_nodes
            next_city = local_opt_tour[next_index]
            used_edge_count = edge_n_used[current_city, next_city]
            if used_edge_count >= 2:
                scaling_factor = np.log(used_edge_count + 1) * 0.5
                updated_edge_distance[current_city, next_city] *= scaling_factor
                updated_edge_distance[next_city, current_city] *= scaling_factor
            else:
                decay_factor = np.exp(-0.1 * used_edge_count)
                updated_edge_distance[current_city, next_city] *= decay_factor
                updated_edge_distance[next_city, current_city] *= decay_factor
            edge_quality = edge_distance[current_city, next_city] / (used_edge_count + 1)
            updated_edge_distance[current_city, next_city] += edge_quality
            updated_edge_distance[next_city, current_city] += edge_quality
    return updated_edge_distance
```

Heuristic 2: Best improvement heuristic discovered for TSP-GLS with size 1000.

```python
import numpy as np
def adaptive_indicators(distance_matrix):
    num_nodes = distance_matrix.shape[0]
    indicators = np.zeros((num_nodes, num_nodes))
    min_edge = np.full(num_nodes, np.inf)
    min_edge[0] = 0
    visited = np.zeros(num_nodes, dtype=bool)
    total_mst_cost = 0
    for _ in range(num_nodes):
        u = np.argmin(np.where(visited, np.inf, min_edge))
        visited[u] = True
        total_mst_cost += min_edge[u]
        for v in range(num_nodes):
            if not visited[v] and distance_matrix[u, v] < min_edge[v]:
                min_edge[v] = distance_matrix[u, v]
    inverted_distance_matrix = 1 / (distance_matrix + np.eye(num_nodes))
    total_density = np.sum(inverted_distance_matrix, axis=1)
    for i in range(num_nodes):
        for j in range(num_nodes):
            if i != j:
                base_indicator = (total_density[i] * total_density[j]) / (1 + total_density[i] +
    total_density[j])
                edge_cost = distance_matrix[i, j] - (total_mst_cost / (num_nodes - 1))
                cycle_penalty = np.sum((inverted_distance_matrix[i, :] + inverted_distance_matrix[j, :] <
    inverted_distance_matrix[i, j]) * distance_matrix[i, j] * 0.2)
```

```
            indicators[i, j] = max(0, (base_indicator - cycle_penalty) * edge_cost)
    max_indicator = np.max(indicators)
    if max_indicator > 0:
        indicators /= max_indicator
    return indicators
```

Heuristic 3: Best improvement heuristic for TSP-KGLS with size 1000.

```
import numpy as np
def score(item, bins):
    scores = np.zeros_like(bins, dtype=float)
    feasible_bins = bins[bins > item]
    if feasible_bins.size == 0:
        return scores
    max_capacity = np.max(feasible_bins)
    scores[bins == max_capacity] = -np.inf
    remaining_capacity = (feasible_bins - item) / feasible_bins
    item_ratio = item / feasible_bins
    proximity_penalty = np.where(feasible_bins >= item * 0.90, -5, 0) + np.where(feasible_bins < item * 0.80,
      -7, 0)
    underutilization_penalty = -3 * np.maximum(0, item - 0.5 * feasible_bins)
    scores[bins > item] = (remaining_capacity + proximity_penalty + underutilization_penalty - (1 -
      item_ratio) ** 3)
    return scores
```

Heuristic 4: Best heuristic for online BPP with 10000 items and a bin capacity of 500.

```
def compute_edge_scores(distance_matrix, coordinates, demands, capacity):
    import numpy as np
    num_nodes = distance_matrix.shape[0]
    edge_promisingness = np.zeros((num_nodes, num_nodes))
    total_demand = np.sum(demands)
    decay_factor = 0.95
    adaptive_alpha = 1.5
    adaptive_beta = 2.5
    for i in range(num_nodes):
        for j in range(num_nodes):
            if i != j and demands[j] <= capacity:
                distance_score = (1 / (distance_matrix[i, j] + 1e-6)) ** adaptive_beta
                demand_score = demands[j] / total_demand if total_demand > 0 else 0
                pheromone_level = 1.0 / (distance_matrix[i, j] + 1e-6) * decay_factor
                exploration_factor = (1 + demands[j] / capacity)
                edge_promisingness[i, j] = (distance_score ** adaptive_beta) * (demand_score **
      adaptive_alpha) * pheromone_level * exploration_factor
    return edge_promisingness
```

Heuristic 5: Best heuristic for CVRP_ACO with size 200.

```
import numpy as np
def compute_pair(demand, capacity):
    n = demand.shape[0]
    heuristic_matrix = np.zeros((n, n))
    valid_indices = np.where(demand <= capacity)[0]
    for i in valid_indices:
        for j in valid_indices:
            if i != j:
                total_demand = demand[i] + demand[j]
                if total_demand <= capacity:
                    heuristic_matrix[i][j] = capacity - total_demand + min(demand[i], demand[j])
    for i in range(n):
        for j in range(n):
            if i != j:
                single_demand = demand[i]
                if single_demand <= capacity:
                    heuristic_matrix[i][j] = max(heuristic_matrix[i][j], capacity - single_demand + demand[j])
    frequency_count = np.sum(heuristic_matrix > 0, axis=1)
    for i in range(n):
        for j in range(n):
            if i != j and heuristic_matrix[i][j] > 0:
                heuristic_matrix[i][j] -= frequency_count[i] * 0.1
    demand_group = np.digitize(demand, bins=np.linspace(0, capacity, num=5))
    for group in range(1, 5):
        group_indices = np.where(demand_group == group)[0]
        if len(group_indices) > 1:
            for i in group_indices:
                for j in group_indices:
                    if i != j and heuristic_matrix[i][j] > 0:
                        heuristic_matrix[i][j] += 0.05
    for i in range(n):
        for j in range(n):
            if i != j and frequency_count[i] > 1 and frequency_count[j] > 1:
                heuristic_matrix[i][j] -= 0.2 * (frequency_count[i] + frequency_count[j]) / 2
    return heuristic_matrix
```

Heuristic 6: Best heuristic for Offline BPP with 1000 items and a bin capacity of 300.

**Seed Meta-Optimizer Example**

```python
def optimize_algorithm(population, utility, language_model,
    subtask_prompt, subtask):
    expertise = "You are an expert in the domain of designing meta
        optimization strategy and combinatorial optimization problems.
        Your task is to design heuristics that can effectively solve
        optimization problems."
    # Step 1: Select a random solution from the population
    selected_solution = population.get_random_solution(subtask)
    # Step 2: Generate directions for improvement
    direction_prompt = (
        f"Given the following heuristic for subtask: {selected_solution['
            best_sol']} with its idea: {selected_solution['idea']} and
            utility score: {selected_solution['utility']}, "
        "Summarize the key idea from this heuristic, then provide
            several totally different ideas from the given one to design
            improved algorithms with lower utility score. "
        "Provide a single string as the answer, less than 50 words. Your
            response should be formatted as a json structure: "
        "```json\n{{\"insights\":[\"content\",\"content\",\"content\",
            ... ,\"content\"]}}\n```."
    )
    response = language_model.prompt(expertise, direction_prompt,
        temperature=1)
    directions = json.loads(extract_code(response))["insights"]
    # Step 3: Create messages based on generated directions
    message_batch = []
    for direction in directions:
        message = (
            f"Improve the following solution:\n"
            f"```python\n{selected_solution}\n```\n"
            f"You must return an improved solution. Formatted as follows:\
                n{subtask_prompt}\n"
            f"To better solve the problem, you are encouraged to develop
                new solutions based on the direction proposed: {direction
                }. "
            "You will be evaluated based on a score function. The lower
                the score, the better the solution.\n"
            "Your response must firstly provide a summary of the key idea
                 inside a brace and marked as a comment, followed by the
                code implementation. "
            "Be as creative as you can under the constraints."
        )
        message_batch.append(message)
    # Step 4: Generate new solutions using the language model
    responses = language_model.prompt_batch(expertise, message_batch,
        temperature=1)
    new_solutions = extract_code(responses)
    new_ideas = extract_idea(responses)
    # Step 5: Evaluate new solutions
    solutions_with_utilities = [(idea, solution, utility(solution,idea,
        subtask)) for idea, solution in zip(new_ideas, new_solutions)]
    best_idea, best_solution, best_utility = min(
        solutions_with_utilities, key=lambda x: x[2])
    return best_idea, best_solution, best_utility
```

Figure 8: The seed meta-optimizer used for training, which randomly selects previous solutions and generates new directions for improvement.

```python
def optimize_algorithm(population, utility, language_model, subtask_prompt, subtask):#
    Adaptive Genetic Algorithm incorporating selection, crossover, and mutation to enhance
     solution diversity and convergence
    expertise = "You are an expert in the domain of designing meta optimization strategy and
         combinatorial optimization problems. Your task is to design heuristics that can
        effectively solve optimization problems."
    # Step 1: Select top-performing solutions for breeding
    elite_count = max(1, population.get_subtask_size(subtask) // 10) # Top 10% as elites
    elites = [population.get_solution_by_index(subtask, i) for i in range(elite_count)]
    # Step 2: Generate directions for diversity through crossover and mutation
    direction_prompt = ( "Given the top-performing solutions, suggest innovative crossover
        and mutation strategies to create diverse and high-quality offspring. Provide your
        response as a JSON with keys 'crossover_methods' and 'mutation_methods', each
        containing a list of strategies." )
    response = language_model.prompt(expertise, direction_prompt, temperature=0.7)
    directions = json.loads(extract_code(response))
    crossover_methods = directions.get("crossover_methods", [])
    mutation_methods = directions.get("mutation_methods", [])
    # Step 3: Create offspring solutions using the generated strategies
    offspring = []
    for method in crossover_methods:
        for i in range(elite_count):
            parent1 = elites[i]
            parent2 = elites[(i + 1) % elite_count]
            crossover_prompt = ( f"Apply the following crossover strategy to combine these two
                 solutions:\nSolution 1: ```python\n{parent1['best_sol']}\n```\nSolution 2:
                 ```python\n{parent2['best_sol']}\n```\nStrategy: {method}\nProvide the new
                offspring solution as a JSON with keys 'idea' and 'best_sol'." )
            offspring_response = language_model.prompt(expertise, crossover_prompt,
                temperature=0.7)
            offspring_data = json.loads(extract_code(offspring_response))
            offspring.append(offspring_data)
    for method in mutation_methods:
        for elite in elites:
            mutation_prompt = ( f"Apply the following mutation strategy to this solution:\
                nSolution: ```python\n{elite['best_sol']}\n```\nStrategy: {method}\nProvide
                the mutated solution as a JSON with keys 'idea' and 'best_sol'." )
            mutation_response = language_model.prompt(expertise, mutation_prompt, temperature
                =0.7)
            mutation_data = json.loads(extract_code(mutation_response))
            offspring.append(mutation_data)
    # Step 4: Evaluate offspring and select the best
    solutions_with_utilities = [ (child['idea'], child['best_sol'], utility(child['best_sol'
        ], child['idea'], subtask)) for child in offspring ]
    # Include elites to maintain the best solutions
    elite_solutions = [ (elite['idea'], elite['best_sol'], elite['utility']) for elite in
        elites ]
    all_candidates = solutions_with_utilities + elite_solutions
    best_idea, best_solution, best_utility = min(all_candidates, key=lambda x: x[2])
    return best_idea, best_solution, best_utility
```

Figure 9: An example of the meta-optimizer generated by LLM, which employs Genetic Algorithm (GA) to balance exploration and expoitation, similar to previous LLM-EC heuristic-optimizer.

```
# {This algorithm employs Ant Colony Optimization (ACO) principles to simulate the foraging
      process of ants for efficient bin packing solutions, enabling adaptive pheromone
    adjustment and heuristic guidance to balance exploration and exploitation for optimal
    results.}
def optimize_algorithm(population, utility, language_model, subtask_prompt, subtask):
    expertise = "You are an expert in the domain of designing meta optimization strategies
        and combinatorial optimization problems. Your task is to design heuristics that can
         effectively solve optimization problems."
    # Parameters for ACO and solution selection
    ant_count = 10
    elite_count = 3
    population_size = population.get_subtask_size(subtask)
    # Step 1: Select elite solutions based on utility for initial pheromone distribution
    elite_solutions = [population.get_solution_by_index(subtask, i)
        for i in range(min(elite_count, population_size))]
    # Initialize pheromone levels for directions based on elite solutions
    pheromone_levels = {sol['best_sol']: 1.0 for sol in elite_solutions}
    # Step 2: Generate solution directions using Ant Colony Optimization principles
    direction_prompts = []
    for _ in range(ant_count):
        direction_prompt = f"Using the ACO principles for the task '{subtask}', generate a
            new solution direction that addresses the bin packing problem. Consider the
            current elite solutions: {[sol['best_sol'] for sol in elite_solutions]} And
            ensure that the output follows this format: {subtask_prompt}. Provide a summary
            comment of the key idea in braces."
        direction_prompts.append(direction_prompt)
    # Step 3: Get new directions from the language model
    responses = language_model.prompt_batch(expertise, direction_prompts, temperature=0.7)
    new_directions = extract_code(responses)
    # Step 4: Evaluate new solutions
    solutions_with_utilities = []
    for direction in new_directions:
        try:
            # Evaluate the new solution's utility
            score = utility(direction, "Derived from ACO strategy", subtask)
            solutions_with_utilities.append((direction, score))
            # Update pheromone based on the quality of the direction
            pheromone_levels[direction] = pheromone_levels.get(direction, 1.0) + 2.0 / (score
                + 1e-6)
        except Exception as e:
            continue # Skip if utility evaluation fails
    # Step 5: Select the best new solution based on its utility score
    if not solutions_with_utilities:
        # Fallback to the best existing solution if no new solutions are valid
        best_existing = population.get_solution_by_index(subtask, 0)
        return best_existing.get('idea'), best_existing.get('best_sol'), best_existing.get('
            utility')
    best_direction, best_utility = min(solutions_with_utilities, key=lambda x: x[1])
    return "Ant Colony optimized direction", best_direction, best_utility
```

Figure 10: An example of the LLM-generated meta-optimizer that utilizes Ant Colony Optimization (ACO) as its underlying mechanism.

Particle Swarm Optimization (PSO)

```python
def optimize_algorithm(population, utility, language_model, subtask_prompt, subtask):
    expertise = "You are an expert in ..."
    # Parameters for PSO
    particle_count = 15
    iterations = 10
    inertia_weight = 0.5 # Controls exploration versus exploitation
    cognitive_param = 0.8 # Personal attraction/learning factor
    social_param = 1.2 # Societal attraction/learning factor
    # Step 1: Initialize particles with random solutions in the population
    particles=[population.get_random_solution(subtask) for _ in range(particle_count)]
    best_personal_solutions = particles.copy()
    global_best_solution = min(particles, key=lambda x: utility(x['best_sol'], "Initial PSO"
        , subtask)) # Initialize global best solution
    # Step 2: Iterate through the PSO process
    for _ in range(iterations):
        for particle in particles:
            current_score = utility(particle['best_sol'], "PSO iteration", subtask)
            # Update personal best if current score is better
            if current_score < utility(best_personal_solutions[particles.index(particle)]['
                best_sol'], "PSO personal best", subtask):
                best_personal_solutions[particles.index(particle)] = particle
            # Update global best if current score is better
            if current_score < utility(global_best_solution['best_sol'], "PSO global best",
                subtask):
                global_best_solution = particle
        # Step 3: Generate new directions using PSO influences
        new_directions = []
        for particle in particles: # Compute new velocities and positions for PSO
            r1, r2 = np.random.rand(), np.random.rand()
            cognitive_velocity = cognitive_param * r1 * (best_personal_solutions[particles.
                index(particle)]['best_sol'] - particle['best_sol'])
            social_velocity = social_param * r2 * (global_best_solution['best_sol'] - particle
                ['best_sol'])
            new_position = particle['best_sol'] + inertia_weight * (cognitive_velocity +
                social_velocity)
            prompt = f"Using PSO principles, generate a new solution for the task '{subtask}'
                based on the solution '{new_position}'. Ensure to follow this format: {
                subtask_prompt}. Provide a summary in braces."
            new_directions.append(prompt)
        # Step 4: Get new solutions from the language model
        responses = language_model.prompt_batch(expertise, new_directions, temperature=0.7)
        generated_solutions = extract_code(responses)
        # Step 5: Evaluate new solutions
        for new_solution in generated_solutions:
            try:
                score = utility(new_solution, "Derived from PSO", subtask)
                if score < utility(global_best_solution['best_sol'], "Final global best",
                    subtask):
                    global_best_solution = {'best_sol': new_solution, 'utility': score}
            except Exception:
                continue # Skip errors in utility evaluation
    return "Particle Swarm optimized direction", global_best_solution['best_sol'],
        global_best_solution['utility']
```

Figure 11: An example of the meta-optimizer generated by LLM, which employs Particle Swarm Optimization (PSO) to balance exploration and expoitation.

**Combination of Adaptive Simulated Annealing with Differential Evolution**

```python
def optimize_algorithm(population, utility, language_model, subtask_prompt, subtask):
    swarm_size = 5
    parent_solutions = [population.get_random_solution(subtask) for _ in range(swarm_size)]
    current_utilities = [parent['utility'] for parent in parent_solutions]
    initial_temperature = 1.0
    cooling_rate = 0.95
    iterations = 5
    best_solution = None
    best_utility = np.inf
    temperature = initial_temperature
    for iteration in range(iterations):
        # Dynamic temperature adjustment based on utility improvements
        temperature = max(initial_temperature * (cooling_rate ** iteration), 0.01) # Ensure
            temperature doesn't go below a threshold
        initial_prompts = f"Using the following solutions: {[sol['best_sol'] for sol in
            parent_solutions]} with utilities: {current_utilities}, devise advanced
            enhancement strategies by blending Adaptive Simulated Annealing and Differential
            Evolution for the task: {subtask}. Return strategies in JSON format: ```json\n
            {{\"strategies\": [\"strategy1\", \"strategy2\", ..., \"strategyN\"]}}```."
        response = language_model.prompt("You are an expert optimization strategist.",
            initial_prompts, temperature=0.7)
        potential_strategies = json.loads(extract_code(response))["strategies"]
        for strategy in potential_strategies:# Apply each strategy to parent solutions
            modified_solutions = []
            for parent in parent_solutions:
                message = f"Implement the strategy: {strategy} on the parent solution: ```
                    python\n{parent['best_sol']}\n```. Generate a modified solution adhering
                    to {subtask_prompt}. Focus on creating diverse and high-quality outputs."
                response = language_model.prompt("You are an expert optimization strategist.",
                    message, temperature=0.7)
                new_solution = extract_code(response)
                new_utility = utility(new_solution, "Optimized by Adaptive Simulated Annealing
                    and DE", subtask)
                if new_utility < best_utility:
                    best_utility = new_utility
                    best_solution = new_solution
                # Explore new areas based on probability
                elif np.random.rand() < np.exp((utility(parent['best_sol'], 'Original', subtask
                    ) - new_utility) / temperature):
                    modified_solutions.append(new_solution)
        # Optional: Implement differential evolution by combining best solutions
        if modified_solutions:
            best_of_modifications = min(modified_solutions, key=lambda s: utility(s, "Modified
                ", subtask))
            new_utility = utility(best_of_modifications, "From Differential Evolution",
                subtask)
            if new_utility < best_utility:
                best_utility = new_utility
                best_solution = best_of_modifications
    best_idea = 'Implemented a hybrid approach of Adaptive Simulated Annealing and
        Differential Evolution for enhanced search efficiency.'
    return best_idea, best_solution, best_utility
```

Figure 12: An example of the meta-optimizer generated by LLM, which combines Adaptive Simulated Annealing with Differential Evolution to explore solution space while refining candidates.

**Tabu Search**

```python
def optimize_algorithm(population, utility, language_model,
    subtask_prompt, subtask):
    # This algorithm utilizes Tabu Search to dynamically explore and
        adapt solutions, avoiding cycling while maximizing job
        performance and minimizing makespan.
    selected_solution = population.get_random_solution(subtask)
    best_solution, best_utility = selected_solution['best_sol'],
        selected_solution['utility']
    tabu_list = set()
    iterations = 0
    max_iterations = 5
    while iterations < max_iterations:
        # Generate candidate solutions
        candidates = []
        for _ in range(3): # Generate 3 candidate solutions
            direction_prompt = (
                f"Given the heuristic: {selected_solution['best_sol']}
                    with its idea: "
                f"{selected_solution['idea']} and utility score: {
                    selected_solution['utility']}, "
                "Generate a modified or new solution. Format as: "
                f"`{subtask_prompt}`. Ensure it is innovative and consider
                     new job scheduling metrics."
            )
            response = language_model.prompt("You are an expert in
                optimization.", direction_prompt, temperature=0.7)
            candidate_code = extract_code(response)
            candidates.append(candidate_code)
        # Evaluate candidates
        candidate_utilities = []
        for candidate in candidates:
            candidate_idea = extract_idea(candidate)
            candidate_utility = utility(candidate_code, candidate_idea,
                subtask)
            candidate_utilities.append((candidate_idea, candidate,
                candidate_utility))
        # Apply Tabu Search logic
        feasible_candidates = [c for c in candidate_utilities if c[1]
            not in tabu_list]
        if feasible_candidates:
            best_candidate = min(feasible_candidates, key=lambda x: x[2])
            best_candidate_idea, best_candidate_solution,
                best_candidate_utility = best_candidate
            # Update best solution if found a better one
            if best_candidate_utility < best_utility:
                best_solution, best_utility = best_candidate_solution,
                    best_candidate_utility
                selected_solution['idea'], selected_solution['best_sol'],
                    selected_solution['utility'] = best_candidate_idea,
                    best_candidate_solution, best_candidate_utility
            # Update Tabu list
            tabu_list.add(best_candidate_solution)
            if len(tabu_list) > 10: # Maintain fixed size
                tabu_list.pop() # Remove the oldest entry
        iterations += 1
    return selected_solution['idea'], best_solution, best_utility
```

Figure 13: An example of the meta-optimizer generated by LLM, which employs Tabu Search to dynamically explore and adapt solutions.

**A Neighborhood Search-like Strategy**

```python
def optimize_algorithm(population, utility, language_model,
    subtask_prompt, subtask):
    # This approach utilizes Variable Neighborhood Search to explore
        different solution neighborhoods dynamically to enhance
        optimization performance.

    def generate_neighborhood(solution):
        # Generate modified solutions (neighborhoods) by tweaking routes
            based on the original solution
        neighborhoods = []
        # Simple route perturbation: swap two routes or modify delivery
            sequences
        for i in range(len(solution['best_sol'])):
            if isinstance(solution['best_sol'][i], list) and len(solution
                ['best_sol'][i]) > 1:
                # Swap two locations in the route
                new_solution = solution['best_sol'][:]
                new_solution[i] = new_solution[i][:2] + new_solution[i
                    ][0:2] # Swap first two for simplicity
                neighborhoods.append({"best_sol": new_solution, "utility":
                    utility(new_solution, subtask)})
        return neighborhoods

    selected_solution = population.get_random_solution(subtask)
    neighborhoods = generate_neighborhood(selected_solution)

    # Collect insights to improve solutions based on the generated
        neighborhoods
    message_batch = []
    for neighbor in neighborhoods:
        message = (
            f"Improve the following solution:\n"
            f"```python\n{neighbor}\n```\n"
            f"You must return an improved solution. Formatted as follows:\
                n{subtask_prompt}\n"
            "To better solve the problem, consider how the neighborhood
                structure changes the optimization landscape. "
        )
        message_batch.append(message)

    responses = language_model.prompt_batch("You are an expert in
        optimizations.", message_batch, temperature=0.7)
    new_solutions = extract_code(responses)
    new_ideas = extract_idea(responses)

    # Evaluate the new solutions and select the best one
    solutions_with_utilities = [(idea, solution, utility(solution, idea,
        subtask)) for idea, solution in zip(new_ideas, new_solutions)]
    best_idea, best_solution, best_utility = min(
        solutions_with_utilities, key=lambda x: x[2])

    return best_idea, best_solution, best_utility
#"best_sol": new_solution, "utility": utility(new_solution, task)
```

Figure 14: An example of the meta-optimizer generated by LLM, which utilizes Variable Neighborhood Search to dynamically explore solution neighborhoods to enhance optimization performance.

Adaptive Exploration-Exploitation Strategy with Dynamic and Tabu Mechanisms

```python
def optimize_algorithm(population, utility, language_model, subtask_prompt, subtask):
    elite_count = 4 # Parameter Initialization
    diversity_count = 3
    pheromone_levels = {}
    # Step 1: Select elite and diverse solutions
    population_size = population.get_subtask_size(subtask)
    elite_solutions = [population.get_solution_by_index(subtask, i) for i in range(min(
        elite_count, population_size))]
    diverse_solutions = [population.get_random_solution(subtask) for _ in range(
        diversity_count)]
    selected_solutions = elite_solutions + diverse_solutions
    # Step 2: Generate dynamic exploration insights with adaptive rates
    for solution in selected_solutions:
        temperature = 1 if solution['utility'] > 0 else 0.75 # Dynamic adjustment
        prompt = f"Given the solution '{solution['best_sol']}' with utility score '{solution['
            utility']}', please suggest innovative optimization strategies that could enhance
            this code. Return your recommendations in JSON format: '''json {{\"insights\":[\"
            content\",\"content\",...]}} '''."
        response = language_model.prompt(expertise, prompt, temperature=temperature)
        try:
            insights = json.loads(extract_code(response))["insights"]
            for insight in insights:
                pheromone_levels[insight] = pheromone_levels.get(insight, 1.0) + 1.0
        except (json.JSONDecodeError, KeyError):
            continue
    # Step 3: Rank directions based on pheromone levels (dynamic evaluation)
    sorted_insights = sorted(pheromone_levels.items(), key=lambda x: x[1], reverse=True)
    top_insights = [insight[0] for insight in sorted_insights if insight[1] > 1.0]
    # Step 4: Create batch messages for generating optimized solutions based on directions
    message_batch = []
    for direction in top_insights:
        message = f"Refine the solution for the task '{subtask}' by focusing on this
            optimization approach: {direction}. Consider elite solutions: {[sol['best_sol']
            for sol in elite_solutions]}. Ensure your output adheres to the following format:
            {subtask_prompt}. In addition, provide a summary of changes made."
        message_batch.append(message)
    # Step 5: Generate new solutions using the language model through batch prompts
    responses = language_model.prompt_batch(expertise, message_batch, temperature=0.9)
    # Step 6: Evaluate and keep the best-performing solutions
    solutions_with_utilities = []
    for response in responses:
        try:
            new_solution = extract_code(response)
            important_idea = extract_idea(response)
            score = utility(new_solution, important_idea, subtask)
            pheromone_levels[important_idea] = pheromone_levels.get(important_idea, 1.0) + (2.0
                / (score + 1e-6))
            solutions_with_utilities.append((important_idea, new_solution, score))
        except Exception:
            continue
    if not solutions_with_utilities: # Best Solution Selection
        best_existing = population.get_solution_by_index(subtask, 0)
        return best_existing['idea'], best_existing['best_sol'], best_existing['utility']
    best_idea, best_solution, best_utility = min(solutions_with_utilities, key=lambda x: x[2])
        # Adjustments for ongoing adaptability of pheromones
    for key in pheromone_levels:
        pheromone_levels[key] *= 0.9 # Mild evaporation to allow exploration
    return best_idea, best_solution, best_utility
```

Figure 15: An example of the meta-optimizer generated by LLM, which employs an adaptive exploration-exploitation strategy that combines real-time performance evaluation of solutions with dynamic exploration rates. This approach customizes search focus within a genetic algorithm framework enhanced by adaptive, tabu-like mechanisms for efficient solution refinement, achieving the **best performance** during inference.

## F  BROADER IMPACTS

This work explores a general framework for improving COP heuristics through LLMs. By introducing a meta-optimization structure, our method demonstrates how LLMs can autonomously generate and improve heuristics across diverse problem domains such as TSP, CVRP, and BPP. Potential social impacts of MoH may include: 1) Improved optimization capabilities in practical applications such as logistics, manufacturing, and resource allocation; 2) Bridging AI and Operations Research (OR) by designing a unified framework that benefits both communities, especially when solving problems with larger sizes; 3) Lower barrier to high-quality algorithm design, especially in low-resource or less-studied problem domains where handcrafted heuristics are not readily available. Meanwhile, a potential negative impact of our method lies in the reliance on LLMs, where both training and inference involve substantial token usage. This can lead to increased energy consumption and raise environmental concerns due to the computational resources required.

## G  THE USE OF LARGE LANGUAGE MODELS

In addition to enhancing the paper writing, LLMs serve as a core component of our methodology to generate and refine optimizers, as well as to support downstream heuristic generation. More precisely, LLMs are used to produce and optimize code implementations aimed at developing high-performing heuristics for solving COPs. A detailed workflow of LLM involvement is presented in Section 3.

## H  LICENSES

We list all the used assets and their licenses in Table 18.

Table 18: Used assets and their licenses.

| Type | Asset | License | Usage |
|---|---|---|---|
| Code | Concorde (Applegate et al., 2003) | Available for academic research use | Evaluation |
| | OR-Tools (Furnon & Perron, 2023) | Apache-2.0 license | Evaluation |
| | FunSearch (Romera-Paredes et al., 2024) | MIT License | Evaluation |
| | EoH (Liu et al., 2024a) | MIT License | Evaluation |
| | ReEvo (Ye et al., 2024) | MIT License | Evaluation |
| | HSEvo (Dat et al., 2024) | MIT License | Evaluation |
| | MCTS-AHD (Zheng et al., 2025) | MIT License | Evaluation |
| | POMO (Kwon et al., 2020) | MIT License | Evaluation |
| | LEHD (Luo et al., 2023) | MIT License | Evaluation |
| Dataset | TSPLib (Reinelt, 1991) | Available for any non-commercial use | Evaluation |

