# OpenReview forum: "Generalizable Heuristic Generation Through LLMs with Meta-Optimization"
_ICLR.cc/2026/Conference — ICLR 2026 Poster_

### Official Review · Reviewer_7Ges · 2025-10-30

**Soundness:** 3
**Presentation:** 3
**Contribution:** 3
**Rating:** 4
**Confidence:** 4

**Summary:**

This paper introduces Meta-Optimization of Heuristics (MoH), a new framework that uses large language models (LLMs) to automatically discover effective heuristic optimizers for combinatorial optimization problems (COPs). Unlike prior work that relies on fixed evolutionary computation (EC) strategies and single-task training, MoH operates at the optimizer level. It employs an outer loop for meta-optimization (optimizer design) and an inner loop for heuristic design. The framework uses multi-task training to improve generalization and supports diverse optimizer discovery through (self-)invocation mechanisms within LLMs. Extensive experiments—including TSP, online/offline Bin Packing, and CVRP—demonstrate that MoH outperforms traditional, neural, and recent LLM-based heuristic generation methods in both performance and scalability.

**Strengths:**

- The shift from fixed heuristic-optimizers to directly searching for optimizers automates the design of entire optimization frameworks, not just heuristics. This represents a conceptual advance over state-of-the-art LLM-based automatic algorithm design.
- MoH is benchmarked across multiple tasks and demonstrates state-of-the-art or competitive optimality gaps, especially when generalizing to larger instance sizes.
- Cost analysis—covering LLM requests, tokens, and wall time—helps contextualize practical applicability.

**Weaknesses:**

- Despite noteworthy generalization to larger problem instances and other COPs, the empirical evaluation focuses on classical, well-studied benchmarks (TSP, BPP, CVRP). The approach hasn't been applied to real-world, domain-constrained, or industrial COPs, which may limit its immediate practical impact. It would be interesting to see how it performs on real-world optimization problems that LLMs are unfamiliar with. Author comments on this aspect would be valuable.
- The distinction between meta-optimizer and heuristic-optimizer is confusing, especially since they turn out to be the same in the end.
- The comparaison to other baselines seems not fair, see questions for details. I would like to hear rebuttal from the authors before making the final decision.

**Questions:**

- During inference, you run MoH for 10 iterations and select the best-performing heuristic based on the final results. Is this strategy also applied to other baselines?
- In Table 8, you distinguish between MoH-Train and MoH-Inference, but Table 9 only mentions MoH. Could you clarify this difference?
- For Tables 8 and 9, testing on large instances increases MoH inference time but not the other methods, correct? The same concern applies to token consumption. I suggest clarifying the entire procedure.

---

> ### Author Response · Authors · 2025-11-25
> **Response to Reviewer 7Ges (1/3)**
>
> We sincerely appreciate the reviewer for taking valuable time to review our paper and for providing insightful suggestion. We hope our responses below could adequately address your remaining concerns.
>
> ***W1: It would be interesting to see how it performs on real-world optimization problems that LLMs are unfamiliar with. Author comments on this aspect would be valuable.***
>
> Thank you for your valuable suggestion. In principle, MoH is a generic framework that is applicable to a broad range of optimization tasks, as any such task can be represented through an executable function (heuristic) and evaluated via a corresponding optimization objective (utility function). We clarify that our initial evaluation focused on standard benchmarks (TSP, BPP, CVRP) primarily to align with the experimental settings of prior AHD baselines (e.g., EoH, ReEvo). We agree with you that implementing on more real-world and industrial problems beyond standard benchmarks would strengthen our practical value.
>
> To directly address your concern, we conducted additional experiments on two distinct problems that differ significantly from routing/packing:
> 1. Quadratic Assignment Problem (QAP): a fundamental model for industrial facility layout planning, representing a heavily constrained discrete optimization task.
> 2. Acrobot System: a classic robotics control task (reinforcement learning environment), where MoH must design a control policy (heuristic) to swing a robot up.
>
> We benchmarked MoH against ReEvo using the settings from [1]. As shown below, **MoH consistently outperforms ReEvo in both domains**, achieving a lower average cost on QAP (averaged over 128 instances) and fewer steps to completion on Acrobot (averaged over 100 runs). These results confirm that MoH is not limited to specific problems but **can effectively adapt to diverse real-world COPs, from industrial layout planning to robotic control.** We will include these results in the revised manuscript.
>
> | Task | Domain | Metric | ReEvo | MoH (Ours) |
> | :--- | :--- | :--- | :--- | :--- |
> | **Acrobot** | Robotics Control | Min. Steps $\downarrow$ | 99.65 $\pm$ 37.60 | **88.76 $\pm$ 14.19** |
> | **QAP** | Facility Layout | Cost $\downarrow$ | 5,913,687.95 | **5,833,648.08** |
>
> [1] Liu, Fei, et al. "Llm4ad: A platform for algorithm design with large language model." arXiv preprint arXiv:2412.17287 (2024).
>
> ***W2: The distinction between meta-optimizer and heuristic-optimizer is confusing, especially since they turn out to be the same in the end.***
>
> Thanks for your comment. We would like to clarify that while they share the same functional template, their fundamental difference lies in their **hierarchical roles and optimization targets** within our bilevel optimization framework:
> - **Meta-optimizer** (Outer Loop): Operates at a higher abstraction level. Its goal is to **design heuristic-optimizers**. It takes a population of heuristic-optimizers as input and evolves them to find better optimization strategies.
> - **Heuristic-optimizer** (Inner Loop): Operates at the task level. Its goal is to **solve specific tasks**. It takes a population of heuristics as input and evolves them to minimize the optimality gap for downstream problems (e.g., TSP, BPP).
>
> Despite these distinct roles, we implement both using a unified functional template (see the code snippet below) to enable recursive self-improvement. This allows the same LLM-generated function structure to adapt its behavior based on the specific inputs it receives.
>
> ```python=
> def optimizer(
>     population: dict,
>     utility: callable[[dict], float],
>     language_model: class ’LanguageModel’
>     subtask_prompt: str,
>     subtask: str) -> Tuple[str, str, float]:
>     ... # internal implementation
> ```
> To be specific,
> - When acting as a Heuristic-Optimizer:
>   - `population`: Stores candidate heuristics for each task.
>   - `utility`: Evaluates task performance (e.g., optimality gap, cost).
>   - `subtask`: Specifies the task type and size (e.g., "TSP20").
> - When acting as a Meta-Optimizer:
>   - `population`: Stores candidate heuristic-optimizers.
>   - `utility`: Aggregated performance across all downstream tasks.
>   - `subtask`: Is set to "Optimizer", directing the function to perform meta-level optimization (as shown in Algorithm 1).

---

> ### Author Response · Authors · 2025-11-25
> **Response to Reviewer 7Ges (2/3)**
>
> ***W3 & Q1: The comparaison to other baselines seems not fair. During inference, you run MoH for 10 iterations and select the best-performing heuristic based on the final results. Is this strategy also applied to other baselines?***
>
> Thanks for the question. We appreciate the opportunity to clarify this. Regarding **Q1: Yes, the same selection strategy applies to all baselines.** In Table 1, for every method, we run the training (or optimization) process and select the best-performing heuristic from the final population. As described in Section 4, each baseline is allocated the same budget of 1,000 heuristic evaluations during training.
>
> To further address your comment, instead of selecting only the best-performing heuristic, we conduct five independent runs (on TSP) for each method and report the corresponding "means ± standard errors" in the table below. The results suggest that MoH consistently achieves lower average optimality gap and demonstrates substantially greater stability, particularly on larger problem sizes.
>
> | Constructive  | 20             | 50             | 100            | 200            | 500            | 1000           |
> | ------------- | -------------- | -------------- | -------------- | -------------- | -------------- | -------------- |
> | Funsearch     | 12.062%±0.907% | 16.049%±1.340% | 18.000%±1.487% | 20.507%±1.757% | 21.646%±2.554% | 22.873%±2.976% |
> | EoH           | 10.115%±0.597% | 13.180%±0.851% | 14.485%±1.076% | 15.890%±1.000% | 16.145%±0.672% | 17.224%±0.672% |
> | ReEvo         | 9.519%±0.192%  | 12.317%±0.269% | 13.379%±0.319% | 14.654%±0.230% | 15.412%±0.233% | 16.258%±0.194% |
> | HsEvo         | 10.938%±2.878% | 12.477%±1.982% | 14.105%±1.639% | 16.315%±1.423% | 18.108%±1.964% | 18.963%±2.250% |
> | MCTS-AHD      | 8.407%±1.140%  | 12.545%±1.284% | 14.128%±1.511% | 15.727%±1.753% | 16.813%±1.454% | 17.306%±1.089% |
> | MoH           | **8.599%±1.032%**  | **12.307%±2.019%** | **13.046%±1.540%** | **14.103%±0.476%** | **14.778%±0.469%** | **15.867%±0.511%** |
> | **Improvement**   |                |                |                |                |                |                |
> | EoH-GLS       | 0.000%±0.000%  | 0.000%±0.000%  | 0.051%±0.013%  | 0.426%±0.043%  | 1.927%±0.491%  | 3.323%±0.516%  |
> | HsEvo-GLS     | 0.000%±0.000%  | 0.000%±0.000%  | 0.076%±0.066%  | 0.729%±0.310%  | 1.726%±1.206%  | 3.261%±0.775%  |
> | ReEvo-GLS     | 0.000%±0.000%  | 0.000%±0.000%  | 0.063%±0.027%  | 0.627%±0.340%  | 2.060%±0.444%  | 3.491%±0.665%  |
> | MoH           | **0.000%±0.000%**  | **0.000%±0.000%**  | **0.031%±0.020%**  | **0.345%±0.039%**  | **1.187%±0.280%**  | **1.889%±0.308%**  |
> | ReEvo-KGLS    | 0.000%±0.000%  | 0.000%±0.000%  | 0.005%±0.002%  | 0.223%±0.002%  | 0.981%±0.004%  | 1.616%±0.020%  |
> | HsEvo-KGLS    | 0.000%±0.000%  | 0.000%±0.000%  | 0.006%±0.002%  | 0.228%±0.006%  | 1.003%±0.046%  | 1.667%±0.071%  |
> | MCTS-AHD-KGLS | 0.000%±0.000%  | 0.000%±0.000%  | 0.011%±0.004%  | 0.239%±0.025%  | 0.942%±0.062%  | 1.478%±0.091%  |
> | MoH-KGLS      | **0.000%±0.000%**  | **0.000%±0.000%**  | **0.004%±0.001%**  | **0.200%±0.021%**  | **0.891%±0.053%**  | **1.474%±0.088%**  |
>
>
>
> ***Q2: In Table 8, you distinguish between MoH-Train and MoH-Inference, but Table 9 only mentions MoH. Could you clarify this difference?***
>
> Thank you for pointing this out. Table 8 (currently Table 11 in paper) reports the computational cost for both MoH-Training and MoH-Inference. Table 9 (currently Table 12 in paper) originally included only the MoH-Training results. We have now added the MoH-Inference results as well, as shown below.
>
> | Methods       | Time/mins | Input Tokens | Output Tokens | Total Tokens | LLM requests | Evaluation count | Results-Obj. 20 | Results-Obj. 50 |
> | ------------- | --------- | ------------ | ------------- | ------------ | ------------ | ---------------- | --------------- | --------------- |
> | MoH-Train     | 321.1     | 986216.0     | 740292.3      | 1726508.3    | 1456.7       | 938.3            | 4.831           | 9.262           |
> | MoH-Inference | 107.5     | 479618.0     | 153284.3      | 632902.3     | 330.3        | 307.3            | 4.837           | 9.254           |
> | ReEvo         | 231.2     | 2122802.0    | 677652.0      | 2800454.0    | 1431.0       | 1000.0           | 4.877           | 9.521           |
> | HSEvo         | 350.7     | 2140030.7    | 670590.0      | 2810620.7    | 1076.7       | 1000.0           | 4.977           | 9.536           |
> | MCTS          | 1330.3    | 2601902.0    | 825037.3      | 3426939.3    | 1490.0       | 1000.0           | 4.881           | 9.233           |

---

> > ### Author Response · Authors · 2025-11-25
> > **Response to Reviewer 7Ges (3/3)**
> >
> > ***Q3: For Tables 8 and 9, testing on large instances increases MoH time but not the other methods, correct? The same concern applies to token consumption.***
> >
> > Thanks for the question. We would like to clarify that **runtime increases with problem size for all methods, not just MoH.**
> >
> > **1. Runtime (increases for ALL methods on larger instances):**
> > - We conducted an additional experiment on TSP-GLS-100 and compared it with the TSP-GLS-200 results from the original Table 8 (currently Table 11). As shown in the table below, runtime increases for all methods when scaling up (e.g., EoH rises from 223.2m to 326.4m).
> > - The runtime scales with problem size because the total duration is dominated by the evaluation step (executing the generated heuristics to calculate their scores). Executing a heuristic on a large instance (e.g., 200 nodes) inherently requires more CPU time than on a small one (e.g., 100 nodes), affecting all methods equally.
> >
> > | Methods       | Time of TSP-GLS-100 (mins)| Time of TSP-GLS-200 (mins)|
> > | ------------- | ---------  | ---------  |
> > | EoH           | 223.2      | 326.4      |
> > | HSEvo         | 214.7      | 291.7      |
> > | MoH-Train       | 196.9  |  238.4 |
> > | MoH-Inference       | 46.7  |  62.7 |
> >
> > **2. Token consumption (driven by strategy, not instance size):** Regarding token consumption (Input/Output/Total Tokens) and LLM requests, we clarify that these metrics are primarily driven by the algorithmic design and prompting strategy, rather than the problem instance size.
> >
> > - Independence from problem size: the role of the LLM is to generate heuristic code (algorithmic logic). Thus, token consumption depends mainly on the structural complexity and length of the generated heuristic script, not on the size of the problem instance. As shown in the table below, token consumption actually decreases as the problem size increases, which may suggest that the heuristic generated for TSP200 has a more concise and efficient code structure.
> > - Structural cost differences: As detailed in Section 4, we enforce a strict evaluation budget (capped at ~1,000 evaluations) to ensure a fair comparison. However, the token cost per evaluation varies by method. Different baselines employ distinct prompting mechanisms. For example, ReEvo requires additional tokens for "reflection," whereas MoH allocates tokens for "meta-optimization."
> >
> > | Method   | TSP Size | Input Tokens | Output Tokens | Total Tokens | LLM requests | Evaluation count |
> > | ------------- | --------- | ------------ | ------------- | ------------ | ------------ | ---------------- |
> > | MoH-Train    | 100     | 927537.0     | 589250.3      | 1516787.3    | 1347.7       | 989.3            |
> > | MoH-Train  | 200    | 743837.0     | 458358.7      | 1202195.7    | 1248.3       | 971.0
> > | MoH-Inference | 100 | 526404.3     | 169028.3      | 695432.6     | 311.3        | 282.7 |
> > | MoH Inference | 200      | 398064.3     | 138350.0      | 536414.3     | 276.7        | 240.0            ||
> > | EoH           | 100     | 1282802.0    | 390085.5      | 1672887.5    | 1119.0       | 1001.7           |
> > | EoH           | 200    | 1437973.0    | 446583.7      | 1884556.7    | 1256.0       | 992.0            |
> > | HSEvo         | 100     | 1291765.3    | 297487.0      | 1589252.3    | 691.3        | 1015.0           |
> > | HSEvo         | 200     | 1188412.3    | 290665.0      | 1479077.3    | 679.7        | 1005.0           |
> >
> > Thank you again for the constructive suggestion. We will include these new results in Appendix C.4. of the revised paper.
> >
> > ----
> > We greatly appreciate the reviewer’s insightful review and constructive input. If any elements of our explanation remain ambiguous, we are ready to clarify them thoroughly. We welcome further questions or suggestions and will continue to improve the manuscript.

---

### Official Review · Reviewer_9cf3 · 2025-10-31

**Soundness:** 2
**Presentation:** 2
**Contribution:** 2
**Rating:** 4
**Confidence:** 3

**Summary:**

The paper presents Meta-Optimization of Heuristics (MoH), a framework that uses large language models (LLMs) to generate effective, interpretable heuristics for combinatorial optimization problems (COPs). Unlike traditional methods, which rely on predefined evolutionary computation (EC) heuristics or single-task training, MoH uses meta-learning to automate the design of meta-optimizers, enabling broader heuristic exploration and better generalization to new problems. The authors demonstrate MoH’s superiority over existing LLM-based methods on classic COPs like the Traveling Salesman Problem (TSP) and Bin Packing Problem (BPP).

**Strengths:**

1. MoH introduces the idea of meta-optimization within LLM-based heuristics for combinatorial optimization, addressing key limitations of existing methods like the lack of diversity in heuristic exploration and challenges in generalization.
2. Extensive experiments demonstrate that MoH outperforms both traditional and LLM-based heuristic methods across various settings, showing its ability to tackle problems like TSP and BPP effectively.

**Weaknesses:**

1. While the authors claim that MoH does not incur significant computational overhead, the introduction of a meta-optimization layer adds complexity, which may increase the time and resources required, especially for large problems.
2. Though MoH performs well on classical COPs, its scalability to more complex or non-classical optimization problems (e.g., real-world applications) has not been thoroughly tested.
3. While multi-task learning is a strength, it could also lead to overfitting on the training tasks if not managed properly. The paper doesn't provide a clear strategy for mitigating such risks.
4. While MoH increases the exploration space, there is no detailed analysis of how efficiently it can explore very large or complex search spaces in comparison to simpler heuristics or other optimization techniques.

**Questions:**

1. Please list up and carefully describe any questions and suggestions for the authors. Think of the things where a response from the author can change your opinion, clarify a confusion or address a limitation. This is important for a productive rebuttal and discussion phase with the authors.
2. While MoH performs well on classical COPs, its scalability to more complex, real-world optimization problems (e.g., dynamic environments, non-classical COPs) has not been thoroughly tested. Can you provide any insights into how MoH might adapt to these problems? Have you considered testing MoH on real-world benchmarks or dynamic problem settings?
3. Multi-task learning is a strength of MoH, but it could also lead to overfitting, especially when tasks are not sufficiently diverse or are too similar. The paper does not clarify how overfitting is mitigated during training. Could you elaborate on the strategies used to ensure that the framework generalizes well across tasks? Did you apply any regularization techniques, cross-validation, or other safeguards to address this risk?
4. While MoH expands the heuristic search space, how does it perform when compared to simpler heuristic methods or other optimization techniques, especially in terms of efficiency? Given the large search space, how does MoH ensure it doesn’t waste resources on ineffective explorations? Could you provide a more detailed analysis of MoH’s efficiency in exploring vast search spaces, especially for large and complex problems?

---

> ### Author Response · Authors · 2025-11-25
> **Response to Reviewer 9cf3 (1/3)**
>
> We sincerely appreciate the reviewer for spending valuable time in reviewing our paper and providing detailed comments. we hope that our response below will address your concerns.
>
> ***W1: The introduction of a meta-optimization layer adds complexity, which may increase the time and resources required, especially for large problems.***
>
> Thank you for your comment. We acknowledge that, from a structural perspective, adding a meta-optimization layer introduces an additional level of theoretical complexity compared with baseline methods. However, this layer brings substantial benefits without significantly increasing empirical computational cost.
>
> First, the meta-layer in MoH is lightweight: it does not involve heavy gradient-based meta-training, but simply uses the LLM to propose different heuristic-optimizers. The overall computational cost remains dominated by the inner loop, where heuristics are executed and evaluated on COP instances. This inner-loop cost is inherent to all baselines and increases with problem size. To directly address the reviewer’s concern, we report the computational cost required by all approaches when training on large-scale TSP500 instances. As shown in the table below, MoH requires a computational cost comparable to that of all baselines.
>
> More importantly, the meta-layer significantly enhances the search process. It enables broader and more structured exploration of the heuristic space, leading to stronger generalization across instance sizes and distributions. Empirically, MoH consistently discovers better heuristics than baselines under the same evaluation budget. Thus, the modest structural complexity introduced by the meta-layer yields meaningful practical gains while introducing minimal empirical computational overhead.
>
> Table 1: Computational cost comparison on large-scale TSP instances (200 → 500), showing that time cost increases for all methods as instance size grows.
> | Size          | 200   |                   | 500   |                   |
> | ------------- | ----- | ----------------- | ----- | ----------------- |
> | **TSP-GLS**       | **Time/mins**  | **Evaluation counts** | **Time/mins** | **Evaluation counts** |
> | MoH-Train     | 238.4 | 971.7             | 364.7 | 982.3             |
> | MoH-Inference | 62.7  | 240.3             | 115.2 | 287               |
> | EoH           | 326.4 | 992               | 382.1 | 1000              |
> | HsEvo         | 291.7 | 1005              | 374.5 | 1011.3            |
>
>
> ***W2 & Q2: Though MoH performs well on classical COPs, its scalability to more complex or non-classical optimization problems (e.g., real-world applications) has not been thoroughly tested. Can you provide any insights into how MoH might adapt to these problems?***
>
> Thank you for your comment. In principle, our method can be applied to other heuristic optimization tasks that can be represented as a function and has a corresponding evaluation metric (utility). Although most existing baselines (e.g., EoH, ReEvo) are evaluated on classical CO problems, we agree that assessing performance on more complex tasks would further strengthen our evaluation.
>
> To this end, we conducted additional experiments on the Quadratic Assignment Problem (QAP) and the Acrobot system control task. For Acrobot, the reported metric is the minimum number of steps required to complete the task across 100 randomly initialized conditions. For QAP, the cost is averaged over 128 instances. All other settings follow [1]. The detailed results are provided below, and we will include these additional experiments in the revised version of the paper.
>
> | Task | Domain | Metric | ReEvo | MoH (Ours) |
> | :--- | :--- | :--- | :--- | :--- |
> | **Acrobot** | Robotics Control | Min. Steps $\downarrow$ | 99.65 $\pm$ 37.60 | **88.76 $\pm$ 14.19** |
> | **QAP** | Facility Layout | Cost $\downarrow$ | 5,913,687.95 | **5,833,648.08** |
>
> [1] Liu, Fei, et al. "Llm4ad: A platform for algorithm design with large language model." arXiv preprint arXiv:2412.17287 (2024).

---

> > ### Author Response · Authors · 2025-11-25
> > **Response to Reviewer 9cf3 (2/3)**
> >
> > ***W3 & Q3: While multi-task learning is a strength, it could also lead to overfitting on the training tasks if not managed properly. The paper doesn't provide a clear strategy for mitigating such risks.***
> >
> > Thanks for raising this point. We agree that overfitting can be a concern in classical multi-task learning. However, MoH fundamentally differs from standard parametric MTL models, and its design includes safeguards that naturally mitigate this risk.
> >
> > First, MoH operates at the level of algorithmic code generation of optimizers, rather than parametric model fitting. Unlike conventional MTL approaches that train a model to fit data, MoH uses an LLM to generate and refine algorithms (e.g., optimizers and heuristics) for COPs. This structural difference inherently reduces overfitting: an algorithm must encode broadly valid algorithmic principles in order to achieve consistently strong performance across diverse validation instances. Overly task-specific heuristics fail quickly and are filtered out through the optimization process.
> >
> > Second, our empirical generalization results show that MoH does not generate heuristics that overfit to a single instance size. Across three settings (tables below), the best heuristic obtained at each training size also maintains strong performance when transferred to neighboring sizes. This demonstrates smooth generalization across adjacent scales, without noticeable degradation or overfitting. Furthermore, for tasks with different input distributions, the corresponding cross-distribution generalization results are provided in our response to `Reviewer 9XE1 W2`. These results further support that MoH does not overfit to the training distribution and can adapt well across varying task characteristics. Overall, both the structural properties of MoH and the empirical evidence indicate that overfitting is not a significant problem for our framework.
> >
> > ***Notes on how to read the table:** Each row corresponds to the best heuristic obtained at a given training size, and each column corresponds to the test size on which that heuristic is evaluated. The tables therefore illustrate how well a size-specific heuristic transfers to neighboring sizes. For example, in the TSP-GLS table, the best heuristic trained at size 200 is evaluated on sizes 100, 200, and 500. Its generalization performance on TSP500 is 1.027%, which increases only slightly compared to the in-distribution performance of 0.936%. This indicates that the heuristic maintains strong performance across neighboring sizes, with no significant degradation, suggesting that it does not suffer from overfitting.*
> >
> > | TSP-GLS | 50    | 100   | 200   | 500   | 1000  |
> > | ------- | ----- | ----- | ----- | ----- | ----- |
> > | 20      | 0.000% | \-    | \-    | \-    | \-    |
> > | 50      | 0.000% | 0.020% | \-    | \-    | \-    |
> > | 100     | 0.000% | 0.012% | 0.410% | \-    | \-    |
> > | 200     | \-    | 0.046% | 0.291% | 1.027% | \- |
> > | 500     | \-    | \-    | 0.514% | 0.936% | 1.476% |
> >
> > | TSP-KGLS | 50    | 100   | 200   | 500   | 1000  |
> > | -------- | ----- | ----- | ----- | ----- | ----- |
> > | 20       | 0.000% | \-    | \-    | \-    | \-    |
> > | 50       | 0.000% | 0.003% | \-    | \-    | \-    |
> > | 100      | 0.000% | 0.002% | 0.198% | \-    | \-    |
> > | 200      | \-    | 0.005% | 0.177% | 0.859% | \-    |
> > | 500      | \-    | \-    | 0.194% | 0.805% | 1.365% |

---

> > > ### Author Response · Authors · 2025-11-25
> > > **Response to Reviewer 9cf3 (3/3)**
> > >
> > > ***W4 & Q4: While MoH increases the exploration space, there is no detailed analysis of how efficiently it can explore very large or complex search spaces in comparison to simpler heuristics or other optimization techniques.***
> > >
> > > Thanks for your comment. Directly optimizing heuristic code is inherently complex, especially within an extremely large search space. Traditional heuristic-design methods typically operate within a fixed algorithmic template and can only make limited modifications, such as hyperparameter tuning, local search, or genetic programming. These approaches offer restricted flexibility, produce rigid variants, and often suffer from low executability. In contrast, with the help of LLMs, MoH can generate and refine a wide variety of algorithms, allowing for far more efficient and flexible exploration of the heuristic space than traditional techniques.
> > >
> > > To further investigate this, we conducted additional experiments using OpenTuner, a framework that optimizes heuristics through hyperparameter tuning, and we also applied Genetic Programming (GP) using DEAP on the TSP-GLS-200 setting. The results are shown in the following table. Compared with the LLM-based baseline HsEvo and our MoH, OpenTuner cannot modify code logic and is restricted to a predefined algorithmic structure, which sharply limits the diversity of heuristics it can explore. Genetic Programming also has inherent limitations. Although GP can mutate program structures, the search space is sparse and fragile—small mutations often produce invalid or ineffective programs. Without substantial pre-defined grammars or domain-specific operators, GP struggles to evolve meaningful high-level algorithmic patterns. In contrast, MoH uses LLM to directly revise heuristic algorithmic logic, enabling a substantially richer and more effective search over heuristic strategies.
> > >
> > >
> > > | TSP-GLS-200   | Initial Heuristic | Final Heuristic |
> > > | --------- | ---------- | ------- |
> > > | Opentuner | 4.07%      | 3.21%   |
> > > | Genetic Programming | 4.07%      |0.92% |
> > > | HsEvo     | 4.07%      | 0.33%   |
> > > | MoH       | 4.07%      | **0.29%**   |
> > >
> > > ----
> > > We are grateful for the reviewer’s careful evaluation and valuable feedback. If any portion of our explanation still seems insufficient, we would be pleased to further clarify. We warmly welcome further advices and are dedicated to refining the manuscript.

---

### Official Review · Reviewer_9XE1 · 2025-11-01

**Soundness:** 2
**Presentation:** 2
**Contribution:** 2
**Rating:** 4
**Confidence:** 4

**Summary:**

This paper proposes Meta-Optimization of Heuristics (MoH), a novel framework that uses Large Language Models (LLMs) to generate generalizable heuristics for Combinatorial Optimization Problems (COPs). Unlike prior methods that optimize heuristics directly, MoH operates at the optimizer level. It aims to discover a highly effective "heuristic-optimizer" by meta-learning. This process involves optimizing a meta-prompt that guides an LLM to sample and refine heuristics. The meta-optimizer is trained to maximize a utility function across a diverse set of tasks (e.g., TSP instances of varying sizes), with the goal of producing heuristics that generalize well. The authors evaluate MoH on the Traveling Salesperson Problem (TSP), demonstrating improved performance over several baseline methods.

**Strengths:**

- The core idea of optimizing the optimizer (via a meta-prompt) rather than just the heuristics themselves is a novel and interesting approach to leveraging LLMs in the optimization domain.
- The paper correctly identifies generalizability as a key weakness in existing heuristic generation methods and explicitly designs its utility function to reward performance across different task distributions (i.e., problem sizes).

**Weaknesses:**

- While the method is described with complex terminology, its core mechanism appears to be a sophisticated form of meta-prompt optimization. The "meta-optimizer" is, in essence, a highly-tuned prompt that guides the LLM to sample effective heuristics. This idea, while implemented well, feels intuitive and perhaps more incremental than a fundamental breakthrough, which may limit the paper's conceptual contribution.
- The paper's "generalizability" claim is weak and potentially misleading. Firstly, the framework does not generalize across problem domains; it requires training a new, specialized "meta-optimizer" for each problem class (TSP, BPP, CVRP). Secondly, even within a single problem, the generalization is limited to varying instance sizes from the same data distribution. There is no evidence that the optimizer generalizes to new instances drawn from a different distribution (e.g., from uniformly distributed TSPs to clustered TSPs).
- The experimental comparison to baselines appears to be unfair. MoH's computational cost includes both a training phase (1,000 heuristic evaluations) and a separate, additional "inference stage" to generate the final heuristics. In contrast, baseline methods like EoH are presented as more "online" and may not have this distinct (and costly) inference phase. For a fair comparison, the baselines should be allocated a total computational budget equal to the sum of MoH's training and inference costs. This is particularly concerning given that the performance gains reported in Table 1 are incremental.

**Questions:**

- If the experiment is run multiple times, will it produce a "meta-optimizer" with similar performance, or are the results highly variant? This is a critical point for assessing the method's reliability.
- Why was the utility function weighted by the size of the task? Figure 1 seems to suggest that performance suffers when emphasizing larger instances. What is the performance of the MoH framework when using a uniform weight for all task sizes? What is the performance if the baselines use the weighted utility?
- What is the performance of the final heuristics obtained at the end of the training phase? This would help clarify the exact performance gain and cost attributed to the separate inference stage.
- Could you clarify the practical difference between the heuristic generation strategy used in the MoH inference stage and the strategy used by the baseline EoH?
- What specific heuristic was tested on the TSPLIB benchmark? Was it the single best heuristic from Table 1? If so, from which problem size distribution was this heuristic generated?
- Why are the results for the ReEvo baseline missing in the TSP-GLS case in Table 1?
- The paper requires careful proofreading to correct several typos (e.g., "generats" in line 74, "hsmaller" in line 928).

---

> ### Author Response · Authors · 2025-11-25
> **Response to Reviewer 9XE1 (1/5)**
>
> We sincerely thank the reviewer for taking the time to evaluate our paper and for providing valuable feedback. We hope our responses below adequately address your concerns.
>
> ***W1: While the method is described with complex terminology, its core mechanism appears to be a sophisticated form of meta-prompt optimization. The "meta-optimizer" is, in essence, a highly-tuned prompt that guides the LLM to sample effective heuristics.***
>
> Thank you for the comment. We would like to gently clarify that characterizing MoH as merely a “sophisticated meta-prompt optimization” may not fully capture the fundamental algorithmic shift introduced by our framework. The key distinction lies in moving beyond optimizing with a static strategy toward designing dynamic, executable search strategies. Our focus is on generating algorithmic code via LLM, not on “prompting prompts".
>
> **We first clarify that our meta-optimizer is more than a meta-prompt.** Concretely, meta-prompt optimization focuses on tuning textual inputs to elicit a better response. In contrast, the MoH meta-optimizer functions as an active algorithmic controller. It does not simply "ask" the LLM; it maintains a stateful process that includes population management, iterative code refinement, and feedback loops. These are structured algorithmic components that exist outside the prompt itself. This unique active algorithmic controller is formalized in our bilevel framework. Unlike prior works that rely on a single-level prompt template, MoH introduces an outer loop that dynamically searches for the optimization strategy itself. This allows MoH to explore the space of algorithms rather than just the space of prompts, a direction largely underexplored in LLM-based combinatorial optimization.
>
> Regarding the concern that the meta-optimizer is “highly tuned”: if it were merely a carefully crafted prompt, it would likely be brittle and model-specific. However, Tables 3 and 14 in paper show that our meta-optimizer generalizes well across diverse LLMs (e.g., GPT, DeepSeek, Qwen) without any modification. Furthermore, we tested multiple prompting strategies within MoH, including CoT, Reflexion, and modified system prompts, on the TSP-GLS setting with size 200. The results below, averaged over five runs, further demonstrate that MoH remains effective across these different configurations. **This consistency proves that the performance drives from the robustness of the optimization logic, rather than overfitting to a highly-tuned prompt template.**
>
> | Prompt strategy   | Gap   |
> | --------- | ------------- |
> | MoH (Default)  | 0.345%±0.039% |
> | MoH (CoT)       | 0.378%±0.037% |
> | MoH (Reflexion) | 0.381%±0.027% |
> | MoH (System)    | 0.399%±0.026% |

---

> > ### Author Response · Authors · 2025-11-25
> > **Response to Reviewer 9XE1 (2/5)**
> >
> > ***W2: The paper's "generalizability" claim is weak and potentially misleading. Firstly, the framework does not generalize across problem domains. Secondly, even within a single problem, the generalization is limited to varying instance sizes from the same data distribution.***
> >
> > Thank you for your comment. To clarify, the main focus of this paper is cross-size generalization, as highlighted in the abstract, introduction, and experimental sections. While our framework can, in principle, be extended to support other forms of generalization, doing so would require addressing additional challenges, which we leave for future work. We will make this scope and positioning clearer in the revised version. Below, we address the additional generalization settings raised by the reviewer:
> >
> > **Cross-problem generalization.** We agree that MoH does not directly generalize a heuristic learned for one CO problem to a completely different CO problem. This limitation is also shared by all baselines we compare against. Although our framework introduces a multi-task training mechanism that can, in principle, incorporate multiple problem settings, using a single trained meta-optimizer across distinct CO problems may not yield strong performance, particularly due to the substantial structural differences (e.g., objective, constraint) among CO problems. Nevertheless, we conducted a preliminary experiment in which MoH was jointly trained on TSP-GLS, TSP-KGLS, and BPP-online tasks. The results (mean±standard error across 5 runs) are shown in the following table. We leave further improvements to future work.
> >
> > | Training Task | Gap |
> > | ------ | ---- |
> > | TSP-GLS-Size200 | 0.384%±0.038%  |
> > | TSP-KGLS-Size200 | 0.219%±0.023%  |
> > | BPP-online-5k item, capacity 100 | 0.978%±0.218%  |
> >
> > **Cross-distribution generalization.** We evaluate cross-distribution generalization in Tables 7 and 8 using TSPLIB, whose instances do not follow the uniform distribution used during training. To further clarify this aspect, we conducted two additional experiments to assess cross-distribution generalization of both the heuristics and the meta-optimizer trained by MoH:
> >
> > * (a) Cross-distribution generalization of the heuristics. We evaluate the generalization ability of the heuristics obtained by each approach in Table 1 by changing the evaluation distribution from uniform to clustered.
> >
> > |      | 20 || 50      |         | 100     |         | 200     |         | 500     |         | 1000    |         |
> > | ----- | ------- | ------- | ------- | ------- | ------- | ------- | ------- | ------- | ------- | ------- | ------- | ------- |
> > |      | Obj.↓   | Gap     | Obj.↓   | Gap     | Obj.↓   | Gap     | Obj.↓   | Gap     | Obj.↓   | Gap     | Obj.↓   | Gap     |
> > | Optimal     | 1.818 | \-      | 2.659 | \-      | 3.629 | \-      | 5.151 | \-      | 8.219 | \-      | 11.430 | \-      |
> > | `TSP Constructive Heuristic` |||||||| | |  | | |
> > | Funsearch    | 2.009   | 10.517% | 3.117   | 17.178% | 4.284   | 18.070% | 6.205   | 20.438% | 9.939   | 20.888% | 13.987  | 22.287% |
> > | EoH                        | 2.021   | 11.173% | 3.086   | 16.055% | 4.200   | 15.653% | 6.026   | 16.857% | 9.618   | 16.929% | 13.463  | 17.684% |
> > | ReEvo                      | 2.007   | 10.407% | 3.071   | 15.506% | 4.230   | 16.488% | 6.068   | 17.836% | 9.711   | 18.130% | 13.786  | 20.561% |
> > | HSEvo                      | 1.964   | 8.027%  | 3.005   | 12.899% | 4.125   | 13.570% | 5.950   | 15.380% | 18.103  | 9.715%  | 13.463  | 17.684% |
> > | MCTS                       | 1.995   | 9.694%  | 3.043   | 14.350% | 4.143   | 14.139% | 5.946   | 15.303% | 9.519   | 15.731% | 13.254  | 15.910% |
> > | **MoH**                    | 1.950   | 7.104%  | 2.993   | 12.478% | 4.113   | 13.177% | 5.927   | 14.990% | 9.451   | 14.913% | 13.190  | 15.361% |
> > | `TSP Improvement Heuristic` |||| |||||||||
> > | EoH-GLS| 1.818   | 0.000%  | 2.659   | 0.000%  | 3.631   | 0.032%  | 5.169   | 0.360%  | 8.376   | 1.909%  | 11.829  | 3.492%  |
> > | HSEvo-GLS  | 1.818   | 0.000%  | 2.659   | 0.000%  | 3.630   | 0.028%  | 5.167   | 0.315%  | 8.311   | 1.110%  | 11.721  | 2.552%  |
> > | ReEvo-GLS                  | 1.818   | 0.000%  | 2.659   | 0.000%  | 3.632   | 0.079%  | 5.166   | 0.290%  | 8.314   | 1.143%  | 11.870  | 3.854%  |
> > | **MoH-GLS**                | 1.818   | 0.000%  | 2.659   | 0.000%  | 3.630   | 0.025%  | 5.165   | 0.275%  | 8.297   | 0.936%  | 11.670  | 2.100%  |
> >
> > - (b) Cross-distribution generalization of the meta-optimizer. We evaluate the generalization ability of the meta-optimizer trained on TSP200-Uniform by leveraging it to optimize heuristics for the TSP200-Cluster task. The following table shows the in-distribution (TSP200-Uniform) and cross-distribution (TSP200-Cluster) performance (best in three runs), respectively.
> >
> > | TSP-GLS| TSP200-Uniform| TSP200-Cluster |
> > | --------|------- | ------------- |
> > | ReEvo    |0.331%|   0.290% |
> > |HSEvo| 0.328%|  0.315%  |
> > | **MoH**     |0.291%|  0.268% |

---

> > > ### Author Response · Authors · 2025-11-25
> > > **Response to Reviewer 9XE1 (3/5)**
> > >
> > > ***W3: For a fair comparison, the baselines should be allocated a total computational budget equal to the sum of MoH's training and inference costs.***
> > >
> > > We appreciate the reviewer’s thoughtful comments. To address this concern, we allocated additional evaluation budget to the baselines to ensure a fair comparison. Specifically, we increased their budgets to 1500 heuristic evaluations, matching (or even surpassing) the total number of evaluations used in MoH’s training and inference. We ran each approach five times and collected the mean ± standard error results. It is important to highlight that simply increasing the evaluation budget does not help the baselines. Under the results from a total of 15 runs for each method (5 runs × 3 baselines), only 2 of them had improvement after 1000 evaluations, and all the results did not surpass the corresponding best results reported in Table 1.
> > > As shown in the table below, even with increased evaluation budget, MoH consistently achieves lower optimality gaps across all problem sizes.
> > >
> > > | TSP-GLS          | 20            | 50            | 100           | 200           | 500           | 1000          |
> > > | ---------| ------------- | ------------- | ------------- | ------------- | ------------- | ------------- |
> > > | EoH   |  0.000%±0.000% | 0.000%±0.000% | 0.059%±0.033% | 0.475%±0.122% | 2.091%±0.253% | 3.257%±0.239% |
> > > | HsEvo | 0.000%±0.000% | 0.000%±0.000% | 0.071%±0.057% | 0.693%±0.135% | 1.863%±0.834% | 3.291%±0.867% |
> > > | ReEvo | 0.000%±0.000% | 0.000%±0.000% | 0.069%±0.027% | 0.671%±0.210% | 2.121%±0.729% | 3.560%±0.436% |
> > > | MoH | **0.000%±0.000%**  | **0.000%±0.000%**  | **0.031%±0.020%**  | **0.345%±0.039%**  | **1.187%±0.280%**  | **1.889%±0.308%**  |
> > >
> > > ***Q1: If the experiment is run multiple times, will it produce a "meta-optimizer" with similar performance, or are the results highly variant?***
> > >
> > > We appreciate the reviewer’s concern. To assess stability, we report the mean and standard error over five independent runs on TSP in the table below. The results show that MoH produces highly consistent outcomes across runs and maintains competitive performance, while also exhibiting substantially greater stability than the baselines, especially on larger instance sizes.
> > >
> > > | Constructive  | 20             | 50             | 100            | 200            | 500            | 1000           |
> > > | ------------- | -------------- | -------------- | -------------- | -------------- | -------------- | -------------- |
> > > | Funsearch     | 12.062%±0.907% | 16.049%±1.340% | 18.000%±1.487% | 20.507%±1.757% | 21.646%±2.554% | 22.873%±2.976% |
> > > | EoH           | 10.115%±0.597% | 13.180%±0.851% | 14.485%±1.076% | 15.890%±1.000% | 16.145%±0.672% | 17.224%±0.672% |
> > > | ReEvo         | 9.519%±0.192%  | 12.317%±0.269% | 13.379%±0.319% | 14.654%±0.230% | 15.412%±0.233% | 16.258%±0.194% |
> > > | HsEvo         | 10.938%±2.878% | 12.477%±1.982% | 14.105%±1.639% | 16.315%±1.423% | 18.108%±1.964% | 18.963%±2.250% |
> > > | MCTS-AHD      | 8.407%±1.140%  | 12.545%±1.284% | 14.128%±1.511% | 15.727%±1.753% | 16.813%±1.454% | 17.306%±1.089% |
> > > | MoH           | **8.599%±1.032%**  | **12.307%±2.019%** | **13.046%±1.540%** | **14.103%±0.476%** | **14.778%±0.469%** | **15.867%±0.511%** |
> > > | **Improvement**   |                |                |                |                |                |                |
> > > | EoH-GLS       | 0.000%±0.000%  | 0.000%±0.000%  | 0.051%±0.013%  | 0.426%±0.043%  | 1.927%±0.491%  | 3.323%±0.516%  |
> > > | HsEvo-GLS     | 0.000%±0.000%  | 0.000%±0.000%  | 0.076%±0.066%  | 0.729%±0.310%  | 1.726%±1.206%  | 3.261%±0.775%  |
> > > | ReEvo-GLS     | 0.000%±0.000%  | 0.000%±0.000%  | 0.063%±0.027%  | 0.627%±0.340%  | 2.060%±0.444%  | 3.491%±0.665%  |
> > > | MoH           | **0.000%±0.000%**  | **0.000%±0.000%**  | **0.031%±0.020%**  | **0.345%±0.039%**  | **1.187%±0.280%**  | **1.889%±0.308%**  |
> > > | ReEvo-KGLS    | 0.000%±0.000%  | 0.000%±0.000%  | 0.005%±0.002%  | 0.223%±0.002%  | 0.981%±0.004%  | 1.616%±0.020%  |
> > > | HsEvo-KGLS    | 0.000%±0.000%  | 0.000%±0.000%  | 0.006%±0.002%  | 0.228%±0.006%  | 1.003%±0.046%  | 1.667%±0.071%  |
> > > | MCTS-AHD-KGLS | 0.000%±0.000%  | 0.000%±0.000%  | 0.011%±0.004%  | 0.239%±0.025%  | 0.942%±0.062%  | 1.478%±0.091%  |
> > > | MoH-KGLS      | **0.000%±0.000%**  | **0.000%±0.000%**  | **0.004%±0.001%**  | **0.200%±0.021%**  | **0.891%±0.053%**  | **1.474%±0.088%**  |

---

> > > > ### Author Response · Authors · 2025-11-25
> > > > **Response to Reviewer 9XE1 (4/5)**
> > > >
> > > > ***Q2: Why was the utility function weighted by the size of the task? Figure 1 seems to suggest that performance suffers when emphasizing larger instances. What is the performance of the MoH framework when using a uniform weight for all task sizes? What is the performance if the baselines use the weighted utility?***
> > > >
> > > > Thank you for the thoughtful question. Our rationale for using size-weighted utility is that CO heuristics typically degrade more significantly as problem size increases. Thus, we place greater emphasis on larger instances to encourage the meta-optimizer to discover heuristics that remain effective at scale, an effect that is empirically supported by the table below. This weighting scheme highlights performance on more challenging problems. To address the reviewer’s concern, we also conducted experiments using uniform weights (-uniform) for MoH and size-weighted utility (-weight) for the baselines. Each result is averaged over 5 independent runs. The results show that MoH consistently outperforms the baselines.
> > > >
> > > > | TSP-GLS       | 20            | 50            | 100           | 200           |
> > > > | ------------- | ------------- | ------------- | ------------- | ------------- |
> > > > | MoH-weight    | 0.000%±0.000% | 0.000%±0.000% | 0.031%±0.020% | 0.345%±0.039% |
> > > > | MoH-uniform   | 0.000%±0.000% | 0.000%±0.000% | 0.060%±0.033% | 0.406%±0.052% |
> > > > | HsEvo-weight  | 0.000%±0.000% | 0.002%±0.002% | 0.099%±0.034% | 0.694%±0.132% |
> > > > | HsEvo-uniform | 0.000%±0.000% | 0.000%±0.000% | 0.076%±0.066% | 0.729%±0.310% |
> > > > | ReEvo-weight  | 0.000%±0.000% | 0.000%±0.000% | 0.086%±0.041% | 0.610%±0.363% |
> > > > | ReEvo-uniform | 0.000%±0.000% | 0.000%±0.000% | 0.063%±0.027% | 0.627%±0.340% |
> > > >
> > > > ***Q3: What is the performance of the final heuristics obtained at the end of the training phase? This would help clarify the exact performance gain and cost attributed to the separate inference stage.***
> > > >
> > > > Thank you for your constructive comment. Below we present the performance of the final heuristics obtained at the end of the training phase across different problem settings. Using TSP as an example, we report the evaluation performance of the best heuristic trained on TSP200 when applied to TSP1000, both before and after the inference stage. The results indicate that the inference stage is crucial for further strengthening final performance, especially in challenging generalization settings.
> > > >
> > > > |            | Size  (Train -> Inference)                                            | Before Inference | After Inference   | Relative Improvement |
> > > > | ---------- | -------------------------------------------------- | --------------------- | ------- | ----------- |
> > > > | TSP-constructive   | 200 -> 1000                                          | 15.902%               | 15.049% | 5.364%       |
> > > > | TSP-GLS    | 200 -> 1000                                          | 2.079%                | 1.476%  | 29.004%      |
> > > > | TSP-KGLS   | 200 -> 1000                                          | 0.923%                | 0.805%  | 12.784%      |
> > > > | BPP-online | 1000 item, 100 capacity -> 10000 item, 100 capacity | 0.874%                | 0.374%  | 57.208%     |
> > > >
> > > > ***Q4: Could you clarify the practical difference between the heuristic generation strategy used in the MoH inference stage and the strategy used by the baseline EoH?***
> > > >
> > > > Thank you for the question. In EoH, heuristic generation is driven by a predefined evolutionary computation (EC) optimizer, where the overall optimization pipeline (inc. crossover, mutation, etc.) is manually designed and remains fixed throughout the process. In contrast, MoH does not rely on a fixed optimizer. Instead, the optimization strategies themselves are generated and iteratively evolved by LLMs during the training phase. During inference, MoH applies these discovered optimization strategies (i.e., meta-optimizer) to optimize heuristics for downstream tasks. As shown in Figures 8–15, the strategies explored by MoH can resemble some traditional optimization frameworks (e.g., PSO, ACO, Simulated Annealing) or form novel hybrid strategies. This enables MoH to explore a richer and more adaptive optimization space than EoH, leading to more effective heuristic design.

---

> > > > > ### Author Response · Authors · 2025-11-25
> > > > > **Response to Reviewer 9XE1 (5/5)**
> > > > >
> > > > > ***Q5: What specific heuristic was tested on the TSPLIB benchmark? Was it the single best heuristic from Table 1? If so, from which problem size distribution was this heuristic generated?***
> > > > >
> > > > > Thanks for your question. We evaluate two improvement-heuristic settings on the TSPLIB benchmark: TSP-GLS and TSP-KGLS.
> > > > >
> > > > > * We did not evaluate TSPLIB using a single “best” heuristic from Table 1, because TSPLIB contains instances with diverse sizes and heterogeneous distributions, and using a single heuristic would be suboptimal. Instead, we adopt an instance-level selection strategy. Specifically, for each TSPLIB instance, MoH evaluates all size-specific best heuristics reported in Table 1 (e.g., those trained on sizes 20, 50, 100, 200) and selects the best-performing one for that particular instance. This allows MoH to adapt to the varying characteristics of TSPLIB instances.
> > > > > * All heuristics used for TSPLIB evaluation are generated from the uniform distribution.
> > > > > * To ensure fairness, we apply the same procedure to all baselines: for each TSPLIB instance, we evaluate multiple trained heuristics and report the best-performing one.
> > > > >
> > > > > ***Q6: Why are the results for the ReEvo baseline missing in the TSP-GLS case in Table 1?***
> > > > >
> > > > > ReEvo employs KGLS in their original paper. To provide a comprehensive comparison, we additionally conducted experiments on TSP-GLS for ReEvo, with the average results (mean±std) obtained from 5 independent runs. We will update the results in the revised paper.
> > > > >
> > > > > | TSP-GLS | 20     | 50     | 100    | 200    | 500    | 1000   |
> > > > > | ------- | ------ | ------ | ------ | ------ | ------ | ------ |
> > > > > | ReEvo (best)   | 0.000% | 0.000% | 0.021% | 0.331% | 1.344% | 2.715% |
> > > > > | ReEvo (mean±std)     | 0.000%±0.000%  | 0.000%±0.000%  | 0.063%±0.027%  | 0.627%±0.340%  | 2.060%±0.444%  | 3.491%±0.665%  |
> > > > > | MoH (best)   | 0.000% | 0.000% | 0.012% | 0.291% | 0.936% | 1.476% |
> > > > > | MoH (mean±std)        | 0.000%±0.000%  | 0.000%±0.000% | 0.031%±0.020%  | 0.345%±0.039%  | 1.187%±0.280%  | 1.889%±0.308% |
> > > > >
> > > > >
> > > > > ***Q7: The paper requires careful proofreading to correct several typos (e.g., "generats" in line 74, "hsmaller" in line 928).***
> > > > >
> > > > > Thanks for your detailed review, we have revised them accordingly.
> > > > >
> > > > > ----
> > > > > We sincerely appreciate the reviewer’s thoughtful assessment and constructive comments. If any of our explanation remains unclear, we would be glad to provide additional clarification. We welcome further suggestions and are fully committed to improving our manuscript.

---

> > > > > > ### Comment · Reviewer_9XE1 · 2025-11-26
> > > > > >
> > > > > > Thank you for the detailed rebuttal. The additional experiments, especially those on computational fairness and cross-distribution generalization, demonstrate the effectiveness of the work. I will revise my score accordingly.

---

> > > > > > > ### Author Response · Authors · 2025-11-26
> > > > > > > **Thank you for your acknowledgement!**
> > > > > > >
> > > > > > > Dear Reviewer 9XE1:
> > > > > > >
> > > > > > > Thank you for taking the time to review our paper and for providing thoughtful and comprehensive feedback! We greatly appreciate your recognition of our contributions, and we will carefully incorporate your suggestions into the final version to further strengthen the manuscript.

---

### Official Review · Reviewer_LJhB · 2025-11-01

**Soundness:** 4
**Presentation:** 3
**Contribution:** 4
**Rating:** 8
**Confidence:** 4

**Summary:**

The paper proposes MoH (Meta-Optimization of Heuristics), a two-level LLM-driven framework that searches not just for task heuristics, but for the heuristic-optimizers that generate them. An outer loop uses a meta-optimizer to invoke an LLM and produce candidate heuristic-optimizers, while an inner loop applies each candidate to evolve concrete heuristics for downstream COP tasks. Selection uses utility on validation tasks, and training is multi-task to encourage cross-size generalization. Experiments on TSP and online BPP report state-of-the-art gaps, especially on larger, unseen sizes. Further ablations suggest benefits from maintaining populations and using natural-language idea descriptions.

**Strengths:**

Ablations show benefits from the proposed idea and examine different LLM backends and population sizes.

The paper provides concrete examples/analysis indicating discovered strategies can resemble or hybridize classic metaheuristics.

The paper shows strong empirical results and cross-size generalization on TSP and Online BPP. The proposed MoH often outperforms baselines.

Multi-task training and controlled evaluation budgets are thoughtfully designed to encourage generalization.

**Weaknesses:**

Improvements over strong baselines can be modest in some settings. Further discussion should be included.

According to experimental setups, main tables emphasize best-of-three runs, which can overstate gains versus mean/variance reporting.

There might exist sensitivity to LLM choice and prompts. The robustness under model drift is uncertain.

**Questions:**

Which two main loops make up the MoH framework?

Does the paper claim cross-size generalization on TSP?

Does the method rely on a population of candidates during search?

What is the difference between heuristic-optimizers and meta-optimizers?

---

> ### Author Response · Authors · 2025-11-25
> **Response to Reviewer LJhB (1/3)**
>
> We sincerely appreciate the reviewer for the constructive comments and the supportive assessment of our paper. We hope the responses below address your points effectively.
>
> ***W1: Further discussion on improvements over strong baselines.***
>
> Thanks for the comment. To provide a complete and transparent view, we report the improvement of MoH over the strongest baseline across all experimental settings, summarized in the tables below. Overall, MoH achieves consistent positive improvement across problem types and scales. In settings where the baselines are already extremely strong (TSP-KGLS), the achievable margin is naturally limited, and the observed gains are modest. However, in more challenging settings with multiple sizes, such as online BPP, MoH achieves substantial improvements, demonstrating its advantage when the problem setting is complex or the baseline heuristics are less effective.
>
> **Table 1: Improvement over the SoTA baseline under all the settings in the paper. Here improvement is measured as the relative reduction compared with SOTA for each setting**
>
> | Size             | 20     | 50     | 100      | 200     | 500     | 1000    |
> | ---------------- | ------ | ------ | -------- | ------- | ------- | ------- |
> | TSP-constructive | 0.878% | -0.121% | 5.437%   | 4.411%  | 6.396%  | 6.295%  |
> | TSP-GLS          | 0.000% | 0.000% | 50.000% | 11.280% | 20.408% | 44.511% |
> | TSP-KGLS         | 0.000% | 0.000% | 33.333%  | 13.235% | 7.151%  | 1.872%  |
>
> | Capacity            | 100     | 200 | 300 | 400 | 500  |
> | ------------------- | ------- | ------- | ------- | ------- | -------- |
> | Online BPP-item size 5k  | 27.449% | 28.415% | 36.614% | 45.550% | 24.370%  |
> | Online BPP-item size 10k | 5.046%  | 25.012% | 31.304% | 44.898% | 57.333% |
>
> ***W2: According to experimental setups, main tables emphasize best-of-three runs, which can overstate gains versus mean/variance reporting.***
>
> Thank you for your constructive feedback.
>
> - We first clarify that our initial use of the best-of-three setting followed the experimental setups in EoH, which has been widely adopted in recent literature.
> - Nonetheless, we agree that reporting full statistical metrics provides a more comprehensive evaluation. We therefore conduct five independent runs for each baseline method and report the corresponding **means ± standard errors** below. The results suggest that MoH consistently achieves lower average optimality gap and demonstrates substantially greater stability, particularly on larger problem sizes, compared to the baselines. We believe these more comprehensive results could adequately clear your concern on the robustness of our results.
>
> Thank you again for the suggestion. We have accordingly included them in the revised paper.
>
> | Constructive  | 20             | 50             | 100            | 200            | 500            | 1000           |
> | ------------- | -------------- | -------------- | -------------- | -------------- | -------------- | -------------- |
> | Funsearch     | 12.062%±0.907% | 16.049%±1.340% | 18.000%±1.487% | 20.507%±1.757% | 21.646%±2.554% | 22.873%±2.976% |
> | EoH           | 10.115%±0.597% | 13.180%±0.851% | 14.485%±1.076% | 15.890%±1.000% | 16.145%±0.672% | 17.224%±0.672% |
> | ReEvo         | 9.519%±0.192%  | 12.317%±0.269% | 13.379%±0.319% | 14.654%±0.230% | 15.412%±0.233% | 16.258%±0.194% |
> | HsEvo         | 10.938%±2.878% | 12.477%±1.982% | 14.105%±1.639% | 16.315%±1.423% | 18.108%±1.964% | 18.963%±2.250% |
> | MCTS-AHD      | 8.407%±1.140%  | 12.545%±1.284% | 14.128%±1.511% | 15.727%±1.753% | 16.813%±1.454% | 17.306%±1.089% |
> | MoH           | **8.599%±1.032%**  | **12.307%±2.019%** | **13.046%±1.540%** | **14.103%±0.476%** | **14.778%±0.469%** | **15.867%±0.511%** |
> | **Improvement**   |                |                |                |                |                |                |
> | EoH-GLS       | 0.000%±0.000%  | 0.000%±0.000%  | 0.051%±0.013%  | 0.426%±0.043%  | 1.927%±0.491%  | 3.323%±0.516%  |
> | HsEvo-GLS     | 0.000%±0.000%  | 0.000%±0.000%  | 0.076%±0.066%  | 0.729%±0.310%  | 1.726%±1.206%  | 3.261%±0.775%  |
> | ReEvo-GLS     | 0.000%±0.000%  | 0.000%±0.000%  | 0.063%±0.027%  | 0.627%±0.340%  | 2.060%±0.444%  | 3.491%±0.665%  |
> | MoH           | **0.000%±0.000%**  | **0.000%±0.000%**  | **0.031%±0.020%**  | **0.345%±0.039%**  | **1.187%±0.280%**  | **1.889%±0.308%**  |
> | ReEvo-KGLS    | 0.000%±0.000%  | 0.000%±0.000%  | 0.005%±0.002%  | 0.223%±0.002%  | 0.981%±0.004%  | 1.616%±0.020%  |
> | HsEvo-KGLS    | 0.000%±0.000%  | 0.000%±0.000%  | 0.006%±0.002%  | 0.228%±0.006%  | 1.003%±0.046%  | 1.667%±0.071%  |
> | MCTS-AHD-KGLS | 0.000%±0.000%  | 0.000%±0.000%  | 0.011%±0.004%  | 0.239%±0.025%  | 0.942%±0.062%  | 1.478%±0.091%  |
> | MoH-KGLS      | **0.000%±0.000%**  | **0.000%±0.000%**  | **0.004%±0.001%**  | **0.200%±0.021%**  | **0.891%±0.053%**  | **1.474%±0.088%**  |

---

> > ### Author Response · Authors · 2025-11-25
> > **Response to Reviewer LJhB (2/3)**
> >
> > ***W3: There might exist sensitivity to LLM choice and prompts. The robustness under model drift is uncertain.***
> >
> > We appreciate the reviewer raising this point.
> >
> > * Regarding robustness across LLMs, as presented in Tables 3 and 14, we evaluated MoH across a diverse range of LLMs and baselines. While absolute performance naturally varies between models, MoH consistently maintains superior performance over baselines across all tested models, suggesting that the efficacy is scourced from our design rather than the underlying base model.
> > * To rigorously address your concern regarding prompt sensitivity, we conducted *additional experiments* by combining MoH with various prompt designs, including CoT, Reflexion, and modified system prompts, on the TSP-GLS setting with size 200. The results, averaged over 5 runs, are reported below. As shown, the performance variance is minimal, confirming that MoH remains highly stable across different prompt configurations.
> >
> > | Prompt strategy   | Gap   |
> > | --------- | ------------- |
> > | MoH (Default)  | 0.345%±0.039% |
> > | MoH (CoT)       | 0.378%±0.037% |
> > | MoH (Reflexion) | 0.381%±0.027% |
> > | MoH (System)    | 0.399%±0.026% |
> >
> > ***Q1: Which two main loops make up the MoH framework?***
> >
> > We appreciate the opportunity to clarify the structure of our framework. As detailed in Algorithm 1 and Figure 2, the MoH framework is built upon a bilevel optimization structure consisting of two main loops. **The outer loop** focuses on optimizer design, where the meta-optimizer generates candidate heuristic-optimizers. These candidates are then executed in **the inner loop** for heuristic design, evolving heuristics for specific downstream tasks. The evaluation results from the inner loop flow back to the outer loop, guiding the iterative improvement of the optimization strategies.
> >
> >
> > ***Q2: Does the paper claim cross-size generalization on TSP?***
> >
> > Yes, we focus on cross-size generalization in the main paper. While prior AHD methods often struggle to generalize beyond their training setting, MoH is explicitly designed to overcome this limitation through its bilevel multi-task training framework. We support this claim with comprehensive evidence across different scales (and distributions):
> >
> > - **Cross-size generalization** (Table 1 in paper): MoH trained on small instances (20-200) consistently outperforms baselines and adapts well on unseen large instances (500 and 1000) through a lightweight inference procedure, demonstrating that it captures fundamental problem-solving patterns rather than overfitting.
> > - **Cross-distribution generalization** (Tables 7 and 8 in paper): we validated MoH on TSPLib (real-world instances with diverse distributions) in Appendix C.1. MoH consistently outperforms baselines trained on uniform distributions, confirming its robust generalizability across different node distributions.
> >
> > ***Q3: Does the method rely on a population of candidates during search?***
> >
> > Yes. As illustrated in Figure 2, MoH maintains populations for both heuristic-optimizers (outer loop) and heuristics (inner loop). These populations preserve elite candidates that serve as few-shot examples to guide the LLM toward more refined and effective responses . We note that maintaining candidate populations is a standard practice in LLM-based evolutionary approaches, utilized by baselines such as EoH, ReEvo, and HSEvo. Please refer to Section 3.3 for further details on our population structure and management.

---

> > > ### Author Response · Authors · 2025-11-25
> > > **Response to Reviewer LJhB (3/3)**
> > >
> > > ***Q4: What is the difference between heuristic-optimizers and meta-optimizers?***
> > >
> > > Thanks for your comment. We would like to clarify that while they share the same functional template, their fundamental difference lies in their **hierarchical roles and optimization targets** within our bilevel optimization framework:
> > > - **Meta-optimizer** (Outer Loop): Operates at a higher abstraction level. Its goal is to **design heuristic-optimizers**. It takes a population of heuristic-optimizers as input and evolves them to find better optimization strategies.
> > > - **Heuristic-optimizer** (Inner Loop): Operates at the task level. Its goal is to **solve specific downstream CO tasks**. It takes a population of heuristics as input and evolves them to minimize the optimality gap for downstream problems (e.g., TSP, BPP).
> > >
> > > Despite these distinct roles, we implement both using a unified functional template (see the code snippet below) to enable recursive self-improvement. This allows the same LLM-generated function structure to adapt its behavior based on the specific inputs it receives.
> > >
> > >
> > >
> > > ```python=
> > > def optimizer(
> > >     population: dict,
> > >     utility: callable[[dict], float],
> > >     language_model: class ’LanguageModel’
> > >     subtask_prompt: str,
> > >     subtask: str) -> Tuple[str, str, float]:
> > >     ... # internal implementation
> > > ```
> > > To be specific,
> > > - When acting as a Heuristic-Optimizer:
> > >   - `population`: Stores candidate heuristics for each task.
> > >   - `utility`: Evaluates task performance (e.g., optimality gap, cost).
> > >   - `subtask`: Specifies the task type and size (e.g., "TSP20").
> > > - When acting as a Meta-Optimizer:
> > >   - `population`: Stores candidate heuristic-optimizers.
> > >   - `utility`: Aggregated performance across all downstream tasks.
> > >   - `subtask`: Is set to "Optimizer", directing the function to perform meta-level optimization (as shown in Algorithm 1).
> > >
> > >
> > > ----
> > > Thank you for your careful review and helpful feedback. We truly appreaciate your strong support to our work. Should any aspects of our response still appear unclear, we are happy to further explain. We welcome additional suggestions and remain committed to strengthening the manuscript.

---

### Meta-Review · Area_Chair_BR8N · 2025-12-28

**Summary:**

This submission proposes a meta-learning framework of optimizing the heuristic optimizer in the outer loop while each heuristic optimizer evolves the task-specific heuristic in the inner loop. It obtains strong cross-size generalization compared to traditional evolutionary algorithms and LLM-based heuristic optimizer.

The list of concerns from reviewers is as follows,
1. Clarity: difference between meta-optimization and heuristic optimization, and between meta-optimized heuristic and tuned meta-prompt
2. Limited type of generalization the proposed method addresses
3. Robustness to LLM and heuristic choice
4. Fairness of comparison
5. Performance on non-standard / realworld tasks
6. Evaluation metrics, best-of-3 vs mean/variance
7. Modest improvements over strong baselines in some settings
8. Tradeoff of inference/train runtime

**Reviewer Concerns:**

The authors' rebuttal and additional experiments should have addressed the majority of the concerns. Reviewer 9XE1 acknowledged his/her concerns on computational fairness and cross-distribution generalization were resolved and raised the rating.

Specifically, concern 1, 3, 4, 6, 8 are well addressed.

Re concern 2, the authors clarified that this submission is focused on cross-size generalization. Nonetheless they also showed additional experiments on cross-problem and cross-distribution generalization, providing more comprehensive assessment to the proposed method.

Re concern 5, the authors provided additional experiments on Quadratic Assignment Problem (QAP) and the Acrobot system control task.

Re concern 7, the authors provide the full results across all experimental settings. They explained the improvement is limited for some settings where the baseline are already extremely strong (TSP-KGLS), while in more challenging settings MoH achieves substantial improvements. Nonetheless, in additional ablations and realworld problems provided in the rebuttal, the advantage of MoH is marginal and not statstically significant in some settings.

**Reviewer Scores:**

Reviewer LJhB would keep the rating of 8. Reviewer 9XE1 has raised the rating from 4 to 6. The questions from reviewer 9cf3 (rating 4) and 7Ges (rating 4) have been properly discussed. I do not think the reviewers would lower their rating after the discussion. They might increase their rating or keep as is depending on how they interpret the advantage of the MoH in additional experiments

---

### Decision · Program_Chairs · 2026-01-26

Accept (Poster)